# Human cytomegalovirus infection coopts chromatin organization to diminish TEAD1 transcription factor activity

Khund Sayeed[1†], Sreeja Parameswaran[1†], Matthew J Beucler[2], Lee E Edsall[1], Andrew VonHandorf[1], Audrey Crowther[1,3], Omer A Donmez[1], Matthew R Hass[1], Scott Richards[1], Carmy R Forney[1], Hayley K Hesse[1], Sydney H Jones[1], Katelyn A Dunn[1], Jay Wright[2], Merrin Man Long Leong[4], Laura A Murray-Nerger[4,5,6], Vijay Yechoor[7], Ben E Gewurz[4], Kenneth M Kaufman[1,8,9], John B Harley[9], Bo Zhao[4], William E Miller[2], Leah C Kottyan[1,8,10]*, Matthew T Weirauch[1,8,10,11,12]*

[1]Center for Autoimmune Genomics and Etiology, Division of Human Genetics, Cincinnati Children's Hospital Medical Center, Cincinnati, United States; [2]Department of Molecular Genetics, Biochemistry and Microbiology, University of Cincinnati, Cincinnati, United States; [3]Immunology Graduate Program, University of Cincinnati College of Medicine, Cincinnati, United States; [4]Department of Medicine, Division of Infectious Diseases, Brigham and Women's Hospital, Harvard Medical School, Boston, United States; [5]Department of Microbiology, Harvard Program in Virology, Harvard Medical School, Boston, United States; [6]Center for Integrated Solutions to Infectious Diseases, Broad Institute of Harvard and MIT, Cambridge, United States; [7]Department of Medicine, University of Pittsburgh School of Medicine, Pittsburgh, United States; [8]Department of Pediatrics, University of Cincinnati College of Medicine, Cincinnati, United States; [9]Research Service, Cincinnati VA Medical Center, Cincinnati, United States; [10]Division of Allergy and Immunology, Cincinnati Children's Hospital Medical Center, Cincinnati, United States; [11]Division of Biomedical Informatics, Cincinnati Children's Hospital Medical Center, Cincinnati, United States; [12]Division of Developmental Biology, Cincinnati Children's Hospital Medical Center, Cincinnati, United States

*For correspondence:
leah.kottyan@cchmc.org (LCK);
Matthew.Weirauch@cchmc.org
(MTW)

†These authors contributed
equally to this work

Competing interest: The authors
declare that no competing
interests exist.

Reviewing Editor: Yamini Dalal,
National Cancer Institute, United
States

## eLife Assessment

This interesting study presents **important** information on how human cytomegalovirus (HCMV) infection disrupts the activity of the TEAD1 transcription factor, leading to widespread chromatin alterations. The strength of evidence in revised manuscript is **convincing**, and includes additional functional data teasing out how TEAD1-driven chromatin changes might influence HCMV replication. This work will be of interest to the virology, chromosome biology and transcriptional co-regulation fields.

**Abstract** Human cytomegalovirus (HCMV) infects up to 80% of the world's population. Here, we show that HCMV infection leads to widespread changes in human chromatin accessibility and chromatin looping, with hundreds of thousands of genomic regions affected 48 hr after infection. Integrative analyses reveal HCMV-induced perturbation of Hippo signaling through drastic reduction of TEAD1 transcription factor activity. We confirm extensive concordant loss of TEAD1 binding,

active H3K27ac histone marks, and chromatin looping interactions upon infection. Our data position TEAD1 at the top of a hierarchy involving multiple altered important developmental pathways. HCMV infection reduces TEAD1 activity through four distinct mechanisms: closing of TEAD1-bound chromatin, reduction of YAP1 and phosphorylated YAP1 levels, reduction of TEAD1 transcript and protein levels, and alteration of *TEAD1* exon 6 usage. Altered TEAD1-based mechanisms are highly enriched at genetic risk loci associated with eye and ear development, providing mechanistic insight into HCMV's established roles in these processes.

## Introduction

Human cytomegalovirus (HCMV) is nearly ubiquitous in the human population. While primary infection of immunocompetent individuals is typically mild or asymptomatic, the virus can cause significant disease in developing neonates and immunocompromised individuals (*Rafailidis et al., 2008*; *Swanson and Schleiss, 2013*). HCMV establishes a lifelong latent infection following primary infection that can periodically reactivate throughout the patient's lifetime, leading to shedding of infectious virus or promotion of virus-associated morbidity in immunocompromised individuals (*Forte et al., 2020*). HCMV can infect a broad range of cells including fibroblasts, epithelial cells, and hematopoietic progenitor stem cells (*Sinzger et al., 2008*). Fibroblasts and epithelial cells are typically supportive of a lytic infection, while hematopoietic cells are the reservoir of latent or persistent infection. Fibroblasts have been widely used to study the cellular changes that occur in response to viral infection and for viral propagation in cell culture (*Fortunato, 2021*; *Sinzger et al., 2008*; *Sinzger et al., 1995*). Like all other viruses, HCMV is an obligate intracellular pathogen that relies on the host cell machinery for its replication, assembly, and egress. These complex host–pathogen interactions vary among different cell types, leading to viral infection manifesting itself in primarily a lytic or latent fashion (*Lee and Grey, 2020*; *Rothenburg and Brennan, 2020*). Infection of permissive cells is accompanied by widespread changes in host cell gene expression. Indeed, several previous studies have examined pathways altered by HCMV infection of in vitro cultured cells at the level of gene expression (*Hein and Weissman, 2022*; *Lee and Grey, 2020*; *Li and Kamil, 2016*; *Van Damme et al., 2016*; *van Den Pol et al., 1999*).

Regulation of gene expression involves intricate interplay between the binding of transcription factors (TFs) and cofactors, chromatin accessibility, histone marks, and long-range chromatin looping interactions (*Lambert et al., 2018*). A hallmark feature of viruses is their dependence on the host nuclear environment and host transcriptional processes. For example, some viruses, such as Epstein–Barr virus and Kaposi sarcoma-associated herpesvirus, extensively alter human chromatin architecture in order to replicate (*Campbell et al., 2022*; *Jiang et al., 2017*). HCMV uses host-derived histones to chromatinize its genome for temporal regulation of the Immediate Early protein IE1 (*Zalckvar et al., 2013*). However, the mechanisms affecting HCMV-driven regulation of host gene expression are not well defined, thus limiting our understanding of the complex interplay between virus and host.

To determine the human gene regulatory mechanisms impacted by HCMV, we examined the changes to chromatin accessibility, TF occupancy, chromatin looping, and gene expression resulting from HCMV infection. We show that HCMV infection in human fibroblasts and retinal epithelial cells induces large-scale global changes in the human chromatin landscape. Accessible chromatin regions that closed upon HCMV infection were highly enriched for predicted TEAD sites and depleted of CTCF (CCCTC-binding factor) sites in both human fibroblasts and retinal epithelial cell lines. Chromatin immunoprecipitation followed by sequencing (ChIP-seq) experiments confirm that TEAD1-binding sites are significantly depleted at sites that become inaccessible with HCMV infection, with concomitant reduction of chromatin looping interactions and H3K27ac levels.

TEAD1 is a downstream effector of the Hippo signaling pathway, which regulates cell proliferation and cell fate to control organ growth and regeneration (*Currey et al., 2021*; *Ma et al., 2019*). TEAD1 activity is primarily controlled by its co-activator YAP1 (*Ma et al., 2019*; *Totaro et al., 2018*). We show that HCMV-induced loss of TEAD1 binding is mediated by four distinct mechanisms: (1) Extensive closing of human chromatin that is normally occupied by TEAD1; (2) Reduction of YAP1 and pYAP1 protein levels; (3) Reduction of TEAD1 transcript and protein levels; and (4) Exclusion of TEAD1 exon 6. Consistent with these observations, pathway enrichment analysis of differentially expressed genes upon HCMV infection reveals extensive perturbation of the Hippo/TEAD signaling

pathway. GWAS-based enrichment analysis reveals that these TEAD1-binding loss events are specifically enriched for genetic variants associated with ear and eye development. Collectively, our data provide novel insights into the mechanisms employed by HCMV upon infection of human cells, with important implications in HCMV-induced human growth defects.

## Results

### HCMV infection extensively re-organizes human chromatin upon infection

Several previous studies have reported extensive changes in human gene expression profiles in HCMV-infected cells (*Hertel and Mocarski, 2004*; *McKinney et al., 2014*; *Nightingale et al., 2018*; *Nogalski et al., 2019*; *Oberstein and Shenk, 2017*). However, the regulatory mechanisms underlying these vast changes remain largely unknown. Gene expression programs defined by *cis*-acting DNA elements such as enhancers and promoters typically occur in open or accessible regions of chromatin. We thus sought to determine if HCMV infection leads to genome-wide alterations in chromatin structure. To this end, we used ATAC-seq (*Corces et al., 2017*) to measure HCMV-induced changes to human chromatin accessibility in two widely used HCMV infection models: human fibroblasts (HS68 cells) and retinal epithelial cells (ARPE-19 cells) (*Figure 1A*). ATAC-seq experiments were performed in duplicate in uninfected conditions or 48 hr post-infection (hpi) (*Figure 1B*).

For each cell line, we obtained >200,000 peaks in both uninfected and infected cells, with Fraction of Reads in Peaks (FRiP) scores greater than 0.5 and Transcription Start Sites enrichment scores >30 (average of 35.1) (*Supplementary file 1*), all of which greatly exceed the recommendations set forth by the ENCODE Project consortium (*The ENCODE Project Consortium, 2012*; *Hitz et al., 2023*; *Luo et al., 2020*; *Kagda et al., 2023*). Likewise, our ATAC-seq peaks align strongly with relevant publicly available datasets (*Supplementary file 1*), and we observed very strong agreement between replicates, with samples tightly clustering first by cell type and then by infection status (*Figure 1—figure supplement 1*). Collectively, these results highlight the high quality and internal consistency of our datasets.

We next sought to systematically identify regions of the human genome with differential chromatin accessibility between infected and uninfected cells. To this end, we used DiffBind (*Ross-Innes et al., 2012*) (see Methods), which identified thousands of chromatin regions (ATAC-seq peaks) with statistically significant changes in signal (*Supplementary file 2*). As expected, most accessible chromatin is common to both uninfected and HCMV-infected cells in both cell types – we designate such regions 'unchanged' (*Figure 2A, B*). In fibroblasts, 38,651 peaks were unique to uninfected cells and hence were closed following infection with HCMV (*Figure 2A, C*). Likewise, 49,003 regions were newly accessible in HCMV-infected fibroblasts. We also observed large HCMV-dependent changes to chromatin accessibility in retinal epithelial cells (*Figure 2B, D*). Comparison of differentially accessible chromatin between ARPE and HFF revealed that the vast majority of the HCMV-induced changes are specific to one of the two cell types (*Figure 2—figure supplement 1*). Collectively, these data indicate that HCMV infection of human cells has widespread effects on the human chromatin accessibility landscape.

Regions of open chromatin largely reflect regulatory regions such as enhancers, which can interact with promoters at large genomic distances spanning many megabases through chromatin looping interactions (*Rowley and Corces, 2018*). To measure the impact of HCMV infection on functional chromatin looping across the human genome, we next performed HiChIP with an antibody against H3K27ac in uninfected and infected fibroblasts (see Methods). Analysis of the resulting HiChIP data revealed 143,882 and 97,815 chromatin looping interactions in uninfected and infected cells, respectively (*Supplementary file 3*). QC analyses using HiC-Pro (*Servant et al., 2015*) indicate that the data are of high quality: the final set of unique valid interaction pairs was 67% of the total sequenced pairs for the uninfected cells and 52% for the infected cells. The number of *trans* interactions was 21% of the sequenced pairs for the uninfected cells and 11% for the infected cells (*Supplementary file 4*), similar to or better than the results obtained in the original HiChIP study (*Mumbach et al., 2016*). Peaks were called from the HiChIP data and compared to peaks called from H3K27ac ChIP-seq data (see Methods), revealing between 82% and 92% HiChIP peak overlap with H3K27ac ChIP-seq peaks (*Supplementary file 4*). Collectively, these results indicate that our HiChIP data are of high quality.

**Figure 1.** Experimental design. Schematic overview of the experimental design. (**A**) Human foreskin fibroblasts or human retinal epithelial cells were infected with the TB40/E strain of human cytomegalovirus (HCMV) at a multiplicity of infection of 5 and 10, respectively. Uninfected and HCMV-infected cells were harvested 48 hr post-infection. (**B**) Gene expression, chromatin accessibility, histone marks of active regulatory elements (H3K27ac), transcription factor occupancy (TEAD1 and CTCF), and chromatin looping were measured genome-wide using RNA-seq, ATAC-seq, ChIP-seq, and HiChIP, respectively. Differential analyses were employed to identify HCMV-dependent functional events on a genome-wide scale.

The online version of this article includes the following figure supplement(s) for figure 1:

*Figure 1 continued on next page*

*Figure 1 continued*

**Figure supplement 1.** Principal component analysis of ATAC-seq replicates of uninfected and human cytomegalovirus (HCMV)-infected fibroblasts and retinal epithelial cells.

We next used the HiChIP data to identify HCMV-dependent differential chromatin looping events (see Methods). In total, uninfected cells have 143,882 loops. With HCMV infection, 90,198 of these loops are lost, and 44,045 new loops are gained (*Supplementary file 3*). Because the number of altered loops was large, we repeated loop calling and differential analysis with FDR values less than 0.05, 0.01, and 0.001 (*Supplementary file 3*). For all three cutoffs, the percentage of loops specific to an infection state was very similar. We also randomly downsampled the number of input pairs used for calling loops to verify that our results were not due to a difference in read depth (*Supplementary file 3*). For the three smaller subsets of data, the number of loops specific to an infection state only changed slightly. The full quantification of each chromatin looping event and comparisons of events between conditions are provided in *Supplementary file 6*.

Next, we examined HCMV-altered regions of chromatin accessibility for enriched transcription factor-binding site motifs (see Methods). We use unchanged accessibility regions as a baseline for motif enrichment analysis, in order to identify motifs specific to chromatin regions that are uniquely accessible in either uninfected or infected cells. In human fibroblasts, all three categories of regions (unchanged, closed with infection, open with infection) were highly enriched for AP-1 motifs (*Figure 3A*, purple dots), reflecting the important role played by this TF family in virtually every cell type regardless of infection status (*Lee et al., 1987*; *Wolf et al., 2023*). Intriguingly, we found substantial differences in motif enrichment for other TF families in HCMV-altered chromatin regions. In regions closed upon HCMV infection, CCAAT-enhancer-binding protein (CEBP), E-box (bound by bHLH TFs), and TEA domain (TEAD)-binding motifs were uniquely enriched (*Figure 3A*, left: blue, orange, and maroon dots along the *Y*-axis, respectively). In contrast, newly closed chromatin in HCMV-infected cells was depleted of CTCF-binding sites (*Figure 3A*, left: dark green dots along the *X*-axis), and newly opened chromatin was enriched slightly more strongly for CTCF sites (*Figure 3A*, right: dark green dots approaching the *Y*-axis).

We next examined TF-binding site motif enrichment in retinal epithelial cells. Similar to human fibroblasts, we found that TEAD motifs were specifically enriched in peaks that were closed with infection (*Figure 3B*, left). Unlike in fibroblasts, we did not identify strong enrichment of CEBP or E-box motifs, presumably due to cell-type differences. In addition to differences in enrichment levels, we also observed substantial and consistent changes in the overall predicted occupancy of TEAD TFs and CTCF in both cell types (*Figure 3C, D*). Taken together, these results indicate that HCMV infection extensively re-organizes the human genome in both fibroblasts and retinal epithelial cells. Further, this re-organization alters the accessibility of predicted binding sites for the TEAD TF family and avoids closing the binding sites of CTCF, in both cell types.

## TEAD1 TF genomic binding is substantially depleted upon HCMV infection

Because chromatin re-organization was more pronounced in fibroblasts (*Figure 2*), we focused on this cell type for subsequent chromatin immunoprecipitation experiments and analyses. We prioritized the two TF families (TEAD and CTCF) with consistent enrichment patterns across cell types within HCMV-dependent chromatin accessibility. We selected TEAD1 to represent the TEAD family because it is the only member whose expression levels are significantly altered by HCMV infection (see Figure 5D). We also examined the genome-wide distribution of H3K27ac, a histone mark correlated with active enhancers.

ChIP-seq experiments for TEAD1, CTCF, and H3K27ac were performed in duplicate in uninfected and HCMV-infected conditions (48 hr post-infection). The resulting data were of high quality, meeting or exceeding ENCODE data quality standards (*Supplementary file 7*), with strong agreement between replicates (*Figure 5—figure supplement 1*). In particular, all H3K27ac datasets had >65,000 peaks and all TEAD1 and CTCF ChIP-seq datasets had >15,000 peaks, with FRiP scores ranging from 0.02 to 0.13 and very strong motif enrichment results (CTCF and TEAD motifs rank #1 in every respective experiment, with p-values $<10^{-3000}$) (*Supplementary file 7*). As predicted, the number of TEAD1 ChIP-seq peaks was substantially diminished in infected cells (75,554 in uninfected compared to 17,567 peaks in

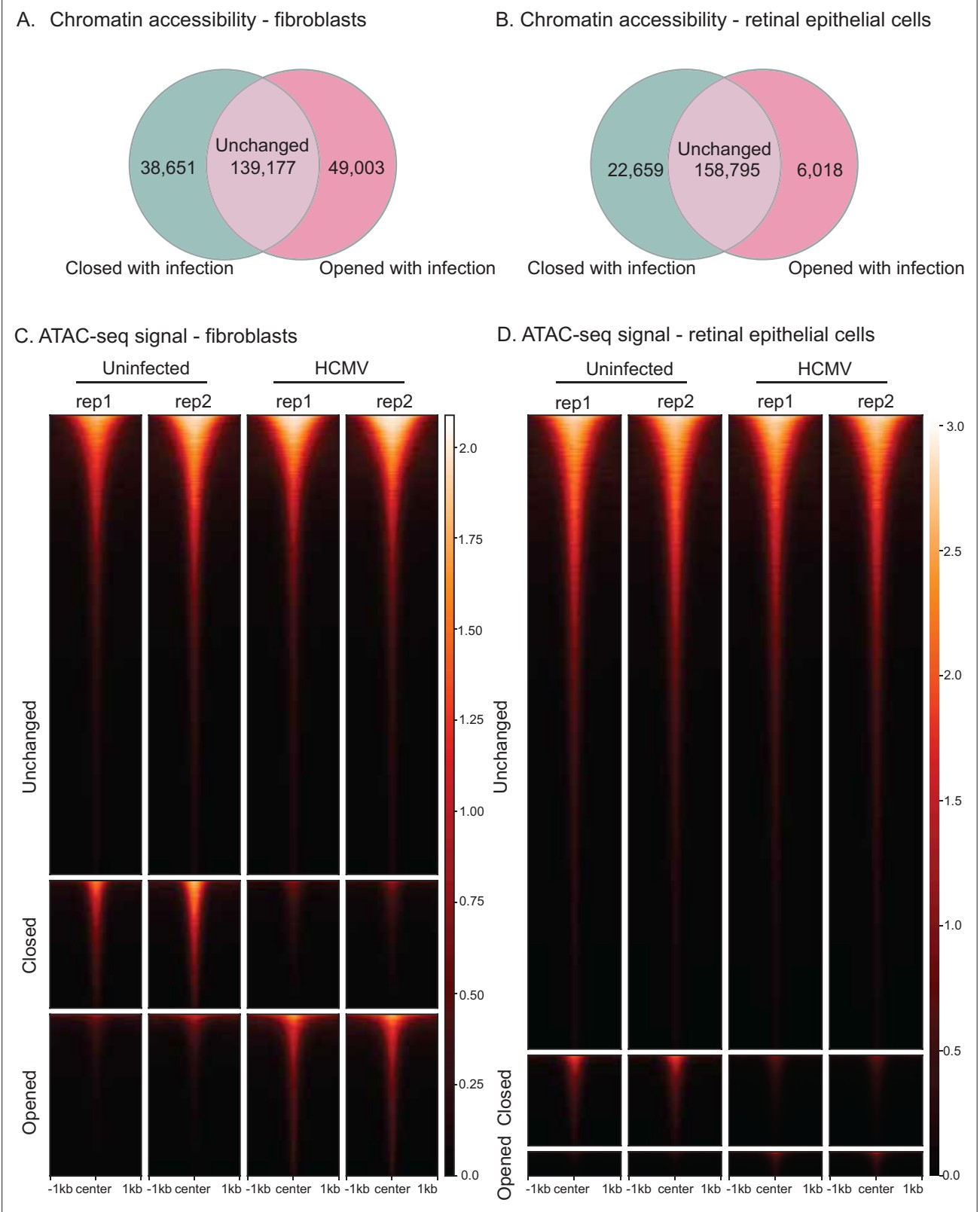

**Figure 2.** Extensive human cytomegalovirus (HCMV)-mediated alterations to human chromatin accessibility. Venn diagram comparing ATAC-seq peaks in uninfected vs. HCMV-infected fibroblasts (**A**) and retinal epithelial cells (**B**). ATAC-seq signal comparison in uninfected and HCMV-infected fibroblasts (**C**) and retinal epithelial cells (**D**).

*Figure 2 continued on next page*

*Figure 2 continued*

The online version of this article includes the following figure supplement(s) for figure 2:

**Figure supplement 1.** Comparison of differentially accessible chromatin across cell types.

**Figure supplement 2.** Comparison of HiChIP signal between replicates of uninfected and human cytomegalovirus (HCMV)-infected cells.

infected cells), along with a substantial drop in H3K27ac peaks (110,308 vs. 66,644) (*Supplementary file 7*), suggesting that loss of TEAD1 binding might have a strong impact on enhancer functionality. In contrast, the total number of CTCF peaks was largely consistent with and without infection (54,697 vs. 57,936 peaks).

To further quantify these changes, we performed differential ChIP-seq peak analysis using the DiffBind software package (*Ross-Innes et al., 2012*) (see Methods). For TEAD1, we identified

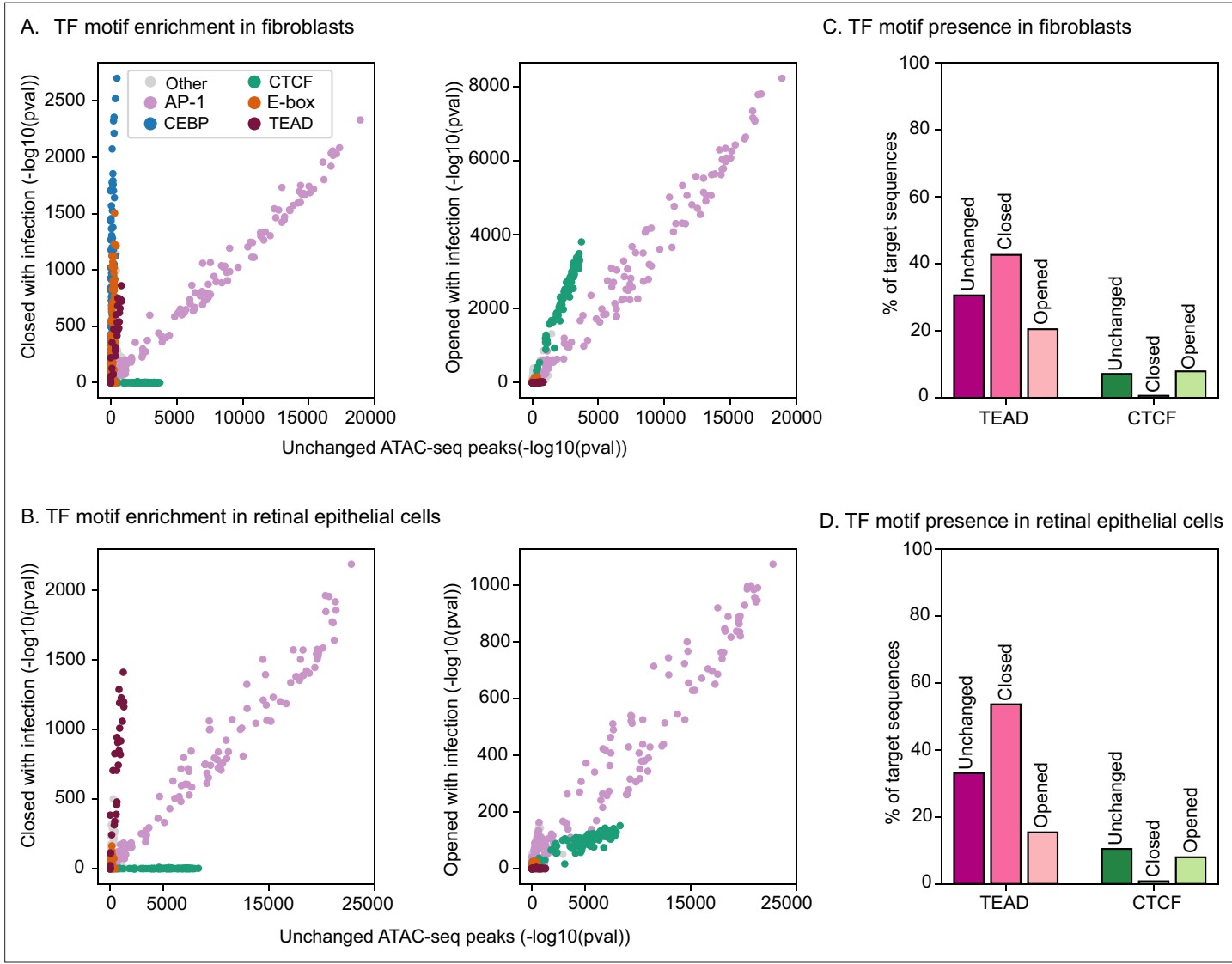

**Figure 3.** Human cytomegalovirus (HCMV) infection alters the accessibility of chromatin containing TEAD DNA-binding motifs and avoids altering CTCF-containing sites. Systematic prediction of human transcription factors (TFs) with HCMV-altered binding. (**A**) Left panel: TF-binding site motif enrichment comparison within ATAC-seq peaks that are unchanged with infection (*X*-axis) vs. peaks that are closed with infection (*Y*-axis). Right panel: same analysis comparing peaks that are unchanged with infection (*X*-axis) and peaks that are opened with infection (*Y*-axis). Each dot represents a human TF-binding site motif. Motifs are color-coded by TF family (see key). (**B**) Same analysis in retinal epithelial cells. (**C**) Percent of predicted binding sites for TEAD and CTCF in ATAC-seq peak regions unchanged with infection and regions closed by HCMV infection. (**D**) Same analysis in retinal epithelial cells.

30,740 peaks specific to the uninfected condition, with only 237 peaks specific to infected cells. Likewise, we identified 40,656 H3K27ac peaks specific to uninfected cells, with 8,542 peaks specific to infected cells (*Supplementary file 8*). As predicted by the motif enrichment analysis of differentially accessible chromatin, regions where the chromatin closed with infection did not contain CTCF ChIP-seq peaks. Instead, we found that losses of TEAD1 occupancy strongly coincide with losses in chromatin accessibility (enrichment: 9.1-fold, adjusted p-value: $1.4 \times 10^{-215}$), losses in chromatin looping events (enrichment: 2.1-fold, adjusted p-value: $3.5 \times 10^{-192}$), and losses in H3K27ac levels (enrichment: 6.1-fold, adjusted p-value: $8.7 \times 10^{-214}$) (*Figure 4A* and *Supplementary file 6*). For example, we observe significant HCMV-induced loss of TEAD1 binding, chromatin accessibility, and H3K27ac levels proximal to the promoters of the Hippo pathway genes *FRMD6* and *RASSF2* (*Figure 4B*), both of which have significantly diminished gene expression subsequent to infection (see next section). In the case of *FRDM6*, which involves a likely enhancer, we also observe loss of a chromatin looping interaction (blue loop) between the promoter (dashed box) and two TEAD1-binding sites lost upon infection (solid box). In the case of *RASSF2*, a TEAD1-binding site within the promoter (solid box) interacts (blue loop) with a downstream enhancer (dashed box). Both the TEAD1-binding site and the interaction are lost upon infection. These examples indicate that loss of TEAD1 binding rather than CTCF binding correlates with loss of chromatin interactions. Collectively, these observations confirm our motif-based predictions that loss of chromatin accessibility co-occurs with loss of TEAD1 binding, in addition to loss of enhancer function (H3K27ac) and enhancer looping interactions (HiChIP).

## HCMV infection induces large-scale changes to human gene expression, including perturbation of the Hippo signaling pathway

Our data indicate substantial changes to human gene regulatory features upon HCMV infection. Therefore, we next systematically assessed HCMV-dependent alterations to human gene expression levels. To this end, RNA was extracted from uninfected and infected human fibroblasts at 48 hr post-infection and analyzed by RNA-seq (see Methods). The resulting data were of high quality (*Supplementary file 9*) and displayed strong agreement between replicates (*Figure 5—figure supplement 2*). As expected, we detected HCMV-encoded genes only in infected cells (*Supplementary file 10*). We identified 2352 differentially regulated human genes (1155 upregulated with infection and 1197 downregulated) at a twofold cutoff with an FDR <0.01 (*Figure 5A* and *Supplementary file 11*).

As expected, pathway enrichment analysis of differentially expressed genes revealed strong enrichment for pathways generally involved in the infection response, including 'viral protein interaction with cytokine and cytokine receptor' and 'human papillomavirus infection' (*Figure 5B*). We also observe strong enrichment for many of the major developmental pathways, including WNT, TGF-beta, and Hippo (*Figure 5B*). Strikingly, many of the genomic regions with extensive (five or more) TEAD1-binding loss events encode major regulators of these pathways, including *WNT5B* and *FZD7* (WNT pathway), *SMAD3*, *TGFB2*, and *TGFBR2* (TGF-beta pathway), and *TEAD1* itself (Hippo pathway), all of which have HCMV-altered expression levels (*Table 1*, *Supplementary file 12*). The extensive loss of TEAD1 binding (with concordant alterations to gene expression levels) suggests that TEAD1 might be a key HCMV-targeted regulator. TEAD TFs are direct effectors of the Hippo signaling pathway (*Currey et al., 2021*), and our data support TEAD being at the top of a hierarchy involving many other key developmental pathways.

Overall, we observe extensive HCMV-induced alterations to Hippo gene expression levels. Of the 163 Hippo pathway genes annotated in the KEGG database, 21 were significantly downregulated (including *TEAD1*) and 20 were upregulated (including *PARD6B*, a negative regulator of Hippo; *Frum et al., 2018*; *Supplementary file 13* and *Figure 5C*). Notably, only TEAD1 was differentially expressed among the four TEAD family members, with a 3.7-fold decrease upon infection (adjusted p-value: $1.9 \times 10^{-29}$) (*Figure 5D* and *Supplementary file 11*). We confirmed alterations in expression levels at the protein level for the classic TEAD1 targets Thrombospondin 1 (THBS1) and Cellular Communication Network Factor 1 (CCN1), both of which were substantially downregulated upon HCMV infection. Specifically, THBS1 protein was not detected in infected cells, while CCN1 was reduced by over threefold (*Figure 5E*).

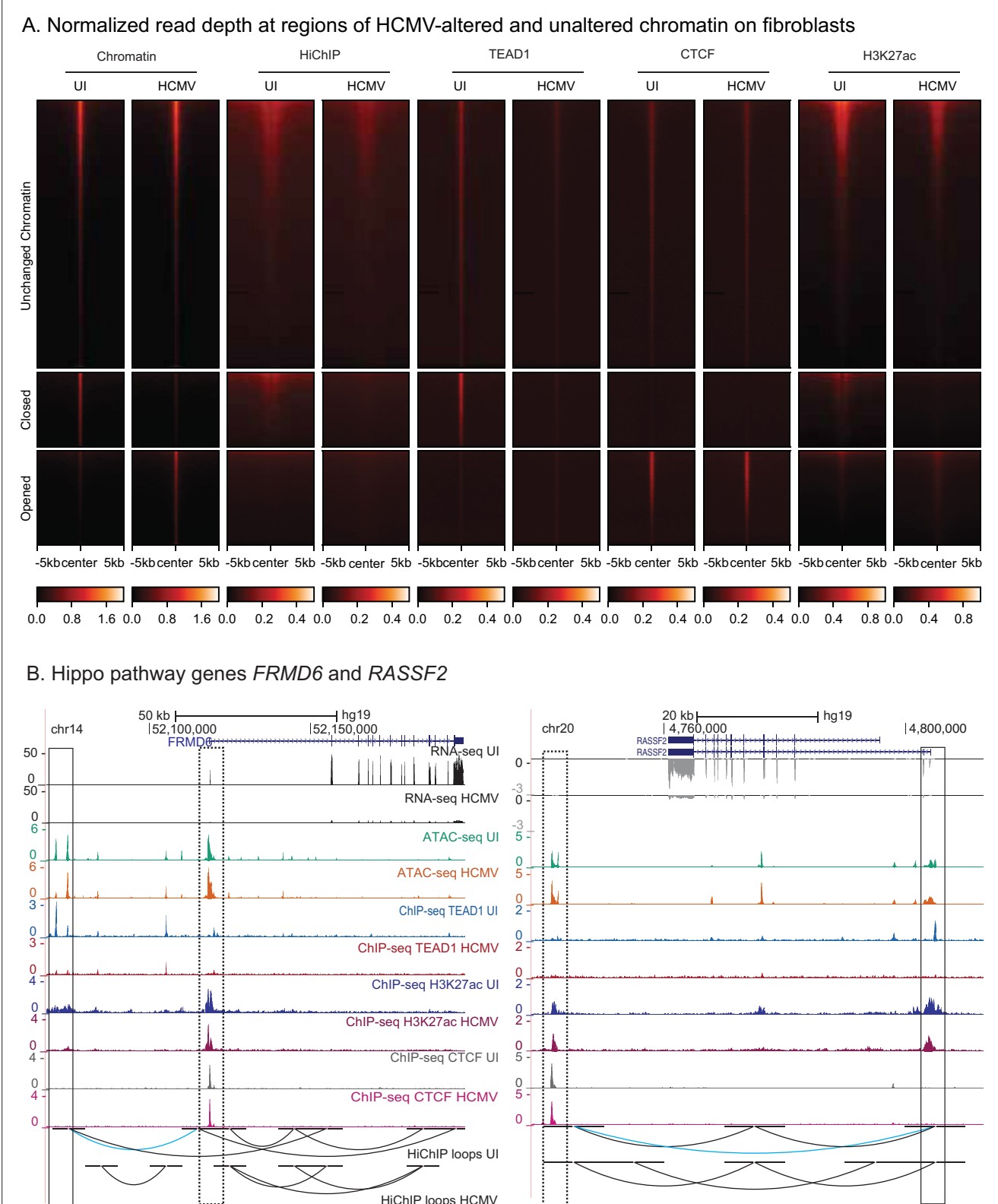

**Figure 4.** Human cytomegalovirus (HCMV) infection leads to widespread coincident loss of chromatin accessibility, TEAD1 binding, H3K27ac marks, and chromatin looping. (**A**) HiChIP and ChIP-seq signal in the context of differentially accessible chromatin regions (ATAC-seq). Regions are split into those containing ATAC-seq signal that is unchanged (top), closed upon infection (middle), or opened upon infection (bottom). The corresponding normalized reads counts are depicted for (left to right): ATAC-seq, HiChIP, and ChIP-seq for TEAD1, CTCF, and H3K27ac. Each row in the heatmaps represents a

*Figure 4 continued on next page*

*Figure 4 continued*

single genomic locus. (**B**) Genome browser images showing depletion of TEAD1, H3K27ac marks, and chromatin looping interactions proximal to Hippo pathway genes *FRMD6* and *RASSF2*. Solid boxes highlight differential TEAD1-binding sites. The *FRMD6* dashed box highlights a promoter, and the *RASSF2* dashed box highlights an enhancer. Chromatin looping interactions lost upon infection are highlighted in blue.

## HCMV infection diminishes TEAD1 TF activity through four distinct mechanisms

We next aimed to further explore the virus-induced mechanisms underlying the reduction in TEAD1 activity. The transcriptional regulatory activity of TEAD1 is primarily controlled by the phosphorylation–dephosphorylation cycle of its co-activator YAP1 (*Ma et al., 2019*; *Totaro et al., 2018*). We thus examined the protein expression levels and phosphorylation status of YAP1 (pYAP1) with and without infection, along with the expression of HCMV proteins IE1/2, TEAD1, and the H3K27ac histone mark in uninfected and infected cells (*Figure 6A*, *Figure 6—figure supplement 1*). Consistent with diminished TEAD1 activity, we observed a reduction in levels of total YAP1 protein (1.8-fold, p < 0.03) and pYAP1 levels (1.5-fold, p < 0.0005) (*Figure 6A*, *Figure 6—figure supplement 1*). It is likely that the reduction of pYAP1 is a direct function of reduced YAP1 expression. These results are consistent with a large-scale HCMV infection proteomics screen performed in human fetal foreskin fibroblasts (*Weekes et al., 2014*). Likewise, TEAD1 protein levels decreased (3.7-fold, p < 0.0009) with HCMV infection (*Figure 6A*, *Figure 6—figure supplement 1*). Consistent with the H3K27ac ChIP-seq results (*Figure 4B*), global H3K27ac was also substantially reduced (1.9-fold, p < 0.19) in infected cells. Since YAP1 activity is primarily controlled by phosphorylation of its serine residues (S127 and S381), which leads to its cytoplasmic translocation (*Pocaterra et al., 2020*), we next assessed the cytoplasmic-nuclear shuttling of pYAP1. These experiments revealed increased YAP1 and pYAP1 in the nuclear fraction of uninfected cells relative to HCMV-infected cells (*Figure 6B*). Taken together, these results indicate that HCMV-induced loss of TEAD1 activity can also partially be accounted for by loss of active YAP1.

Recent reports have shown that alternative splicing of *TEAD1* also regulates TEAD1 activity (*Choi et al., 2022*). We thus used the AltAnalyze software package (*Emig et al., 2010*) to systematically examine alternative splicing changes between HCMV-infected and uninfected cells within our RNA-seq data (see Methods). This analysis identified two differential exon use events within the *TEAD1* gene that are impacted by HCMV infection: (1) skipping of exon 6 with HCMV infection and (2) diminished inclusion of an upstream intronic region just upstream (~90 bases) of exon 5 with HCMV infection (*Figure 6—figure supplement 2*). A recent study carefully examined the functional effect of the alternative exon 6 usage event, concluding that skipping of exon 6 diminishes TEAD1 activity by disrupting the intramolecular interaction between its DNA-binding domain and the YAP1-interacting domain (*Choi et al., 2022*). We confirmed HCMV-induced differential splicing of TEAD1 exon 6 by RT-PCR of total RNA isolated from uninfected and HCMV-infected fibroblasts (*Figure 6*, *Figure 6—figure supplement 2*).

Taken together, our results reveal multiple interrelated mechanisms by which HCMV infection impairs TEAD1 activity (*Figure 6D*): (1) Exclusion of TEAD1 exon 6 (*Figure 6C*); (2) Reduction of *TEAD1* gene and TEAD1 protein levels (*Figures 5D and 6A*); (3) Reduction of YAP1 and pYAP1 protein levels (*Figure 6A*); and (4) Extensive closing of human chromatin that is normally occupied by TEAD1 (*Figure 4*).

## HCMV-induced TEAD1-binding loss coincides with genetic variants associated with ear and eye growth defect phenotypes

Next, we examined the enrichment of GWAS hits within regions of HCMV-induced loss of TEAD1 binding. To this end, we used our RELI tool (*Harley et al., 2018*) to estimate GWAS-associated risk locus enrichment for 1337 diseases and phenotypes. This analysis produced three phenotypes with specific enrichment for TEAD1-binding loss events compared to unchanged TEAD1-binding events (see Methods and *Supplementary file 14*): lobe attachment (*Adhikari et al., 2015*; *Shaffer et al., 2017*), optic cup area (*Bonnemaijer et al., 2019*; *Gharahkhani et al., 2018*; *Springelkamp et al., 2017*; *Springelkamp et al., 2015*), and percent mammographic density (*Lindström et al., 2014*; *Lindström et al., 2011*; *Sieh et al., 2020*). Among these, enrichment replicated in multiple GWAS

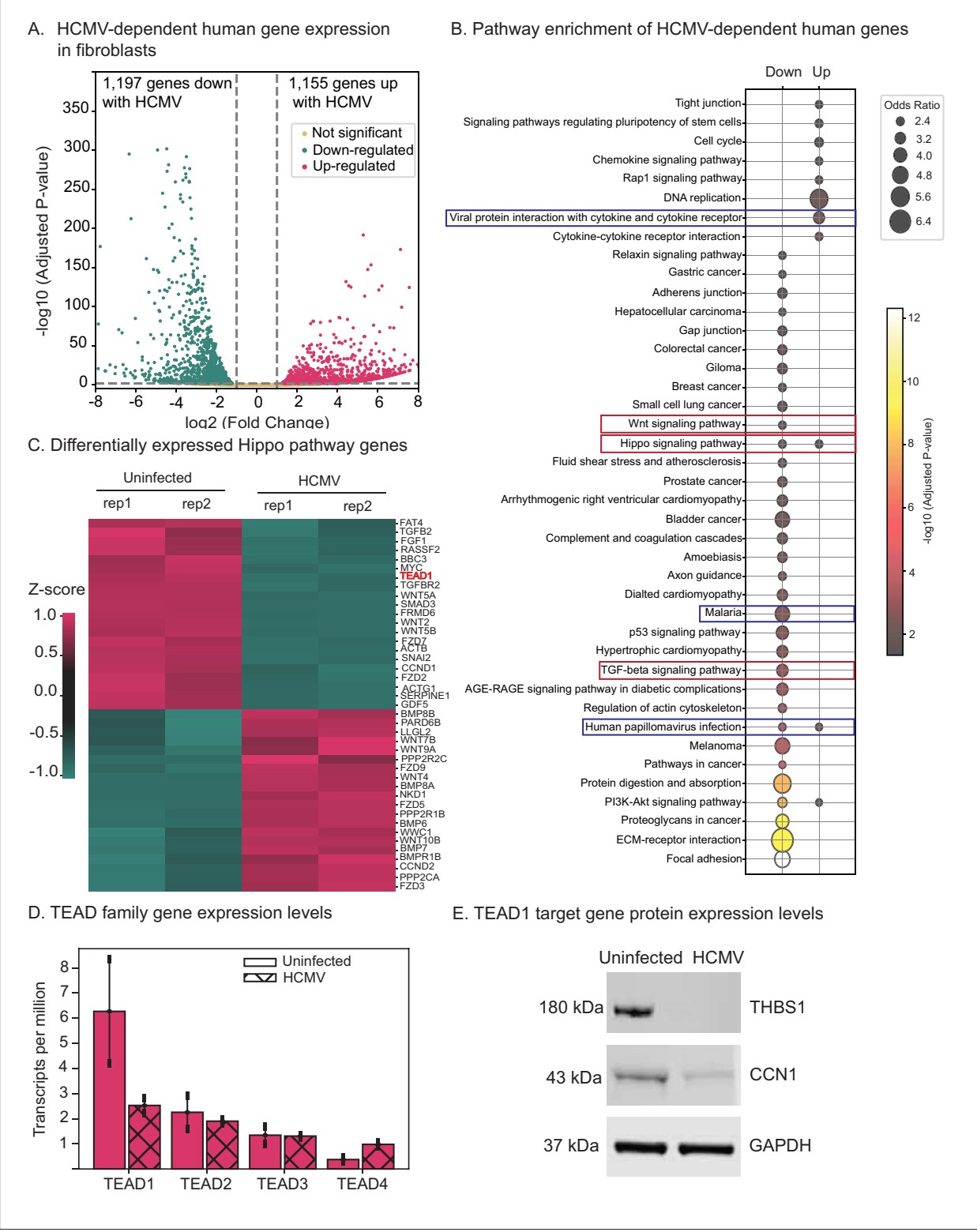

**Figure 5.** Human cytomegalovirus (HCMV) infection alters Hippo signaling gene and protein expression levels. (**A**) Differentially expressed genes in fibroblasts with HCMV infection. (**B**) KEGG pathway enrichment analysis of differentially expressed genes. Pathways relevant to infection are highlighted with blue boxes. Key developmental pathways are highlighted with red boxes. (**C**) Differentially expressed genes within the Hippo pathway. (**D**) Gene expression profiles (transcripts per million [TPM] values) of all four TEAD family transcription factors with and without HCMV infection. (**E**) Western blots of established TEAD1 targets THBS1 and CCN1 using whole-cell lysates of uninfected and HCMV-infected cells. GAPDH is used as a control.

*Figure 5 continued on next page*

*Figure 5 continued*

The online version of this article includes the following source data and figure supplement(s) for figure 5:

**Source data 1.** Raw image data without labels.

**Source data 2.** Raw image data with labels.

**Figure supplement 1.** Heatmap of ChIP-seq peaks for TEAD1, CTCF, and H3K27ac between replicates in uninfected and human cytomegalovirus (HCMV)-infected fibroblasts.

**Figure supplement 2.** Heatmap of gene expression profiles between replicates of uninfected and human cytomegalovirus (HCMV)-infected fibroblasts.

for two phenotypes: lobe attachment and optic cup area (*Figure 6E*). Notably, HCMV infection of fibroblasts has previously been shown to contribute to hearing impairment (*Fowler and Boppana, 2006*; *Goderis et al., 2014*; *Nystad et al., 2008*; *Maheshwari et al., 2024*) and MCMV infection can lead to hearing loss in mice (*Bradford et al., 2015*). Likewise, Hippo signaling plays established roles in ear growth and hearing loss (*Deng et al., 2016*; *Gnedeva et al., 2020*; *Holden and Cunningham, 2018*; *Wang et al., 2022*). Of note, the thrombospondin (*THBS1*) gene, which plays important, well-established roles in hearing loss (*Smeriglio et al., 2019*), has extensive proximal HCMV-induced TEAD1-binding loss and HCMV-induced gene (*Supplementary file 11*) and protein level reduction (*Figure 5E*). Meanwhile, the 'optic cup area' phenotype refers to the overall size of the optic nerve, a phenotype influenced by HCMV infection of the eye (*Ijezie et al., 2023*; *Xu et al., 2020*). Hippo signaling plays well-established roles in eye development (*Moon and Kim, 2018*). Collectively, these results implicate HCMV-induced TEAD1-binding loss in both hearing and eyesight growth defects at multiple specific genomic loci, providing mechanistic insights into the well-established roles of both HCMV and Hippo signaling in these disorders.

## Discussion

HCMV is a leading cause of birth defects in developing fetuses, including ear and eye developmental disorders. Upon infection, HCMV alters the expression of thousands of human genes. A better understanding of the mechanisms underlying these changes is critical for understanding the processes underlying HCMV-associated diseases. In this study, we used several complementary functional genomics approaches along with biochemical validation to show that HCMV employs multiple inter-related molecular mechanisms that diminish the activity of the human TEAD1 TF: closing of chromatin in regions normally bound by TEAD1 (with concomitant reduction of H3K27ac levels and chromatin looping interactions), lowering of TEAD1 protein expression levels, preferential exclusion of TEAD1 exon 6, and lowering of protein expression of the TEAD1 co-activator YAP1. This widespread reduction in TEAD1 activity results in substantial alteration to the expression of many human genes, including targets of Hippo and other key developmental signaling pathways.

The human genome encodes four TEAD TFs (TEAD1–4), all of which are mediators of the Hippo signaling pathway. The Hippo pathway regulates key aspects of development, including control of organ size, stem cell identity, and lineage specification (*Currey et al., 2021*; *Ma et al., 2019*). The activity of TEAD TFs is regulated by the availability of their cofactors YAP1, TAZ (WWTR1), and members of the vestigial-like protein family (VGLL1–4). While YAP1 and TAZ function as transcriptional co-activators of TEAD TFs, VGLL proteins function as co-repressors (*Currey et al., 2021*; *Rausch and Hansen, 2020*; *Yamaguchi, 2020*). The activity of TEAD1 has previously been shown to be affected by differential use of exon 6, with exclusion of exon 6 reducing activity due to an impaired ability to interact with YAP1 (*Choi et al., 2022*). The YAP1–TEAD complex exerts its activity by binding to enhancers of genes involved in extracellular matrix organization, actin cytoskeleton organization, and cell adhesion (*Rausch and Hansen, 2020*; *Stein et al., 2015*), all of which are pathways for which we see significantly altered gene expression levels subsequent to HCMV infection (*Figure 5B*).

A previous report has shown that HCMV infection inhibits the proliferation and invasion of extra-villous cytotrophoblasts (EVT) through mRNA downregulation of core Hippo pathway components, including downregulation of YAP1/TAZ and all four TEAD TFs (*Kong et al., 2021*). Here, we observe a threefold depletion in TEAD1 protein levels and a substantial reduction in YAP1 protein levels, in addition to other mechanisms impacting TEAD1 activity. Additionally, the expression of key TEAD1 targets, including Connective Tissue Growth Factor (CTGF), Fibronectin (FN), α-smooth muscle actin

**Table 1.** Genomic regions with extensive loss of TEAD1-binding events upon human cytomegalovirus (HCMV) infection.

In an unbiased analysis, a 300-kb window was drawn around the transcription start site of each gene with differential expression upon HCMV infection. The number of TEAD1-binding loss events within this window was then counted. All results with five or more TEAD1 loss events are provided in this table. Many of these genes encode members of the Hippo, TGF-beta, and WNT signaling pathways (see text).

| Chr | Start | End | Number of TEAD1 loss events | Gene |
|-----|-------|-----|------------------------------|------|
| chr15 | 67,200,000 | 67,500,000 | 19 | SMAD3 |
| chr14 | 51,800,000 | 52,100,000 | 17 | FRMD6 |
| chr3 | 30,500,000 | 30,800,000 | 14 | TGFBR2 |
| chr11 | 12,500,000 | 12,800,000 | 11 | TEAD1 |
| chr8 | 128,600,000 | 128,900,000 | 11 | MYC |
| chr1 | 218,400,000 | 218,700,000 | 10 | TGFB2 |
| chr12 | 1,600,000 | 1,900,000 | 10 | WNT5B |
| chr19 | 47,600,000 | 47,900,000 | 10 | BBC3 |
| chr5 | 141,900,000 | 142,200,000 | 10 | FGF1 |
| chr11 | 111,500,000 | 111,800,000 | 9 | PPP2R1B |
| chr2 | 202,700,000 | 203,000,000 | 9 | FZD7 |
| chr6 | 7,600,000 | 7,900,000 | 9 | BMP6 |
| chr4 | 126,100,000 | 126,400,000 | 8 | FAT4 |
| chr7 | 100,600,000 | 100,900,000 | 8 | SERPINE1 |
| chr7 | 116,800,000 | 117,100,000 | 8 | WNT2 |
| chr8 | 28,200,000 | 28,500,000 | 8 | FZD3 |
| chr11 | 69,300,000 | 69,600,000 | 7 | CCND1 |
| chr17 | 42,500,000 | 42,800,000 | 7 | FZD2 |
| chr20 | 33,900,000 | 34,200,000 | 7 | GDF5 |
| chr3 | 55,400,000 | 55,700,000 | 7 | WNT5A |
| chr6 | 132,100,000 | 132,400,000 | 7 | CTGF |
| chr17 | 73,400,000 | 73,700,000 | 6 | LLGL2 |
| chr2 | 208,500,000 | 208,800,000 | 6 | FZD5 |
| chr20 | 49,200,000 | 49,500,000 | 6 | PARD6B |
| chr7 | 5,400,000 | 5,700,000 | 6 | ACTB |
| chr1 | 39,800,000 | 40,100,000 | 5 | BMP8A |
| chr17 | 79,300,000 | 79,600,000 | 5 | ACTG1 |
| chr22 | 46,200,000 | 46,500,000 | 5 | WNT7B |
| chr4 | 95,500,000 | 95,800,000 | 5 | BMPR1B |
| chr5 | 167,600,000 | 167,900,000 | 5 | WWC1 |

(ACTA2), and Collagen 1a (COL1A) has all previously been found to be downregulated with HCMV infection of trabecular meshwork cells, leading to an increase in intra-ocular pressure, a leading cause of glaucoma (*Choi et al., 2017*). Although the exact mechanisms seem to differ depending on the cell type, HCMV infection of permissible cells clearly perturbs the Hippo pathway in multiple cellular contexts by diminishing TEAD TF activity.

Recent studies have shown that other viruses perturb the Hippo pathway as part of their replicative cycle (*Wang et al., 2019*). For example, oncogenic viruses, such as Kaposi's sarcoma-associated

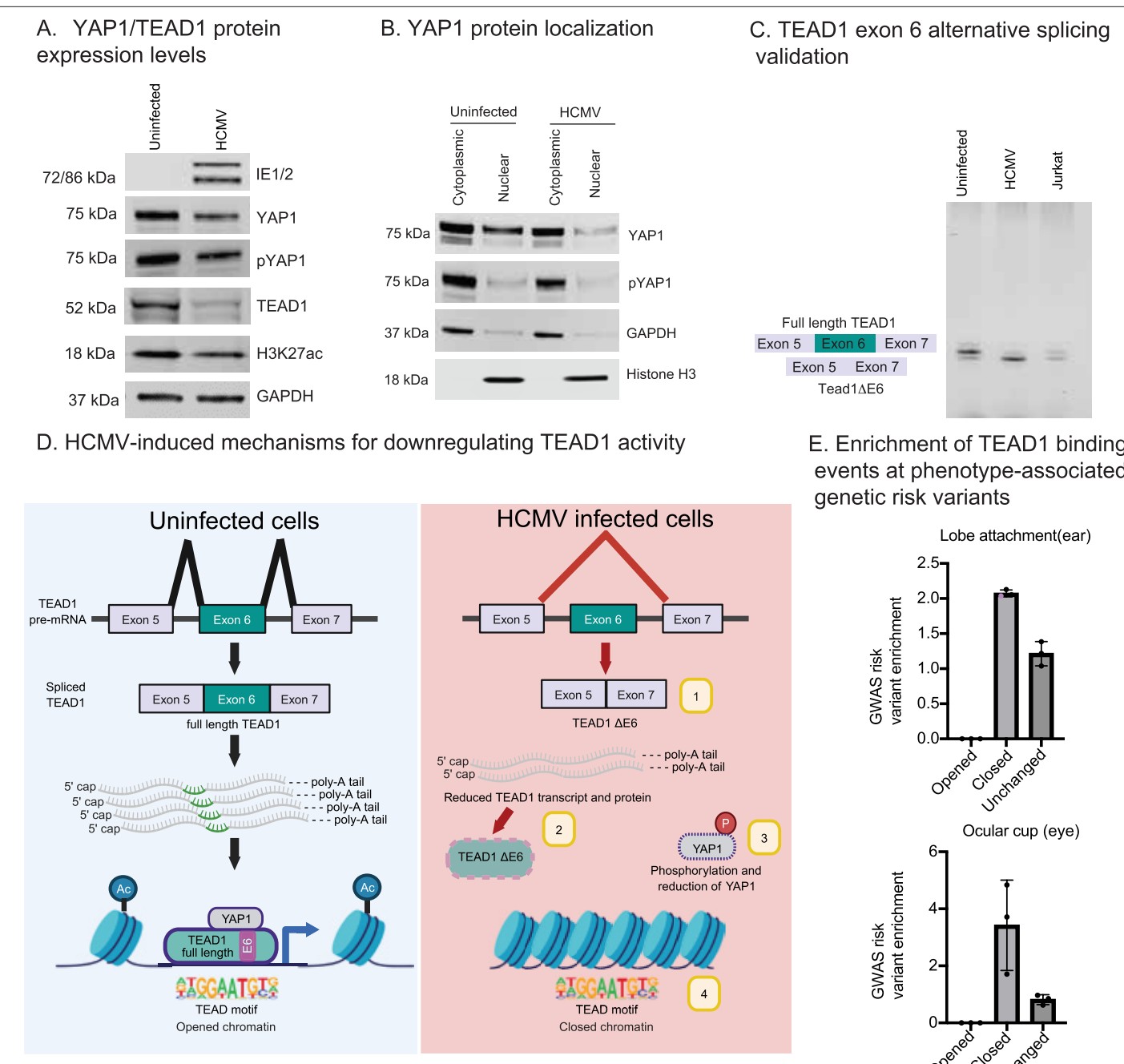

**Figure 6.** Human cytomegalovirus (HCMV) impairs TEAD1 activity through multiple distinct mechanisms. (**A**) Representative Western blots for HCMV proteins IE1/2, YAP1, pYAP1, TEAD1, and the H3K27ac mark from whole-cell lysates of uninfected and HCMV-infected cells. GAPDH was used as a loading control. Additional Western blots (biological triplicates) are provided in **Figure 6—figure supplement 1**, along with quantifications and p-values. (**B**) Western blots using cytosolic and nuclear fractions obtained from uninfected and HCMV-infected cells indicating the localization of YAP1 and pYAP1. GAPDH and Histone H3 were used as controls for cytoplasmic and nuclear fractions, respectively. (**C**) Agarose gel image of RT-PCR products of TEAD1 exon 6 splicing events. The full-length *TEAD1* targeted region is 91 bp long; it is 79 bp without exon 6. Jurkat cells, which have approximately equal expression of TEAD1 with and without exon 6 (**Choi et al., 2022**), were used as a control. (**D**) Model depicting four distinct mechanisms by which HCMV reduces the activity of the TEAD1 transcription factor. (**E**) Enrichment of phenotype-associated genetic variants at HCMV-altered TEAD1-binding events. Enrichment values calculated by the RELI algorithm are presented for TEAD1-binding events for the lobe attachment (**A**) and ocular cup (**B**) phenotypes. Enrichment for TEAD1-binding loss with HCMV infection is statistically significant for each assessment except for the group for lobe attachment (shown in purple). For each bar, dots represent RELI enrichment results from different ancestral groups as defined in the original GWAS studies.

*Figure 6 continued on next page*

*Figure 6 continued*

The online version of this article includes the following source data and figure supplement(s) for figure 6:

**Source data 1.** Raw image files without labels.

**Source data 2.** Raw image files without labels.

**Figure supplement 1.** Western blots of uninfected and human cytomegalovirus (HCMV)-infected fibroblasts in biological triplicates for TEAD1, H3K27ac, YAP1, pYAP1, and HCMV Immediate Early proteins IE1/2.

**Figure supplement 1—source data 1.** Raw image files without labels.

**Figure supplement 1—source data 2.** Raw image files with labels.

**Figure supplement 2.** TEAD1 alternative splicing event summary.

herpesvirus (KSHV) and Human Papillomavirus (HPV), promote dephosphorylation and nuclear translocation of YAP1 (*Liu et al., 2015*). In the case of KSHV, the viral GPCR (vGPCR; ORF74) has been shown to inhibit the Hippo kinases LATS1/2, thereby activating YAP1/TAZ (*Liu et al., 2015*). Likewise, HPV16 oncoprotein E6 induces cell proliferation by dephosphorylating a serine residue of YAP1 (S379) and preventing its degradation by the proteasome complex (*He et al., 2015*). More recently, it has been discovered that Epstein-Barr virus subverts the YAP/TAZ pathway in lytic reactivation in epithelial cells (*Singh et al., 2023*; *Van Sciver et al., 2021*).

The results of this study represent one of likely many examples of a single virus targeting the same host molecule through multiple mechanisms. For example, HPV targets TP53 through independent mechanisms involving both the E6 and E7 proteins (*Fontan et al., 2022*). Likewise, poxviruses target caspase activity through multiple mechanisms (*Nichols et al., 2017*). Given the vast amount of evolutionary time that viruses have at their disposal due to their highly elevated mutation rates, and the reliance of viruses on host-encoded pathways, it is likely that many other viruses also target a single molecule through multiple mechanisms.

Several previous studies have examined changes to human gene expression induced by HCMV infection in a variety of contexts (reviewed in *Martí-Carreras and Maes, 2019*). To date, only two studies have employed more than one type of genome-scale measurement to compare HCMV-infected to uninfected cells. In the first study, multiple single-cell approaches were employed (including scATAC-seq and CITE-seq) to compare NK cells with or without infection, revealing a possible role for AP-1 family TFs (*Rückert et al., 2022*). In the second study, integration of Pol2, H3K27ac, and H3K27me3 ChIP-seq data with ATAC-seq data revealed epigenetic reprogramming of the virus and host genomes in myeloid progenitor cells (*Forte et al., 2021*). Interestingly, this study found only limited changes to chromatin accessibility between infected and uninfected cells, likely due to differences in cell type (Kasumi-3 vs. HS-68 and ARPE-19 in our study), time point (24 vs. 48 hr), infection strength (multiplicity of infection [MOI] of 1 with <50% infection vs. MOI of 5 and 10 with >90% infection), and replication efficiency (~30 vs.~250 viral genomes per cell). To our knowledge, our study is the first to employ a systematic approach to identify important host transcriptional regulators through unbiased analysis of virus-altered chromatin accessibility regions, with subsequent ChIP-seq-based validation. Future studies will be needed to systematically identify important host-encoded regulators in the context of other cell types and other viruses.

HCMV infects up to 86% of the world's population (*Zuhair et al., 2019*). For most, this infection is largely benign. However, infection can cause complications in newborn infants and the immunocompromised. Our comprehensive analyses reveal that HCMV targets the Hippo signaling pathway through TEAD1, setting off a cascade of effects on other key developmental pathways. Importantly, these TEAD1-binding loss events are highly enriched for genetic variants associated with eye and ear phenotypes, providing possible mechanistic insights into the well-established role of HCMV infection in eye and ear disorders. Collectively, the results of our study offer new avenues to investigate HCMV-dependent mechanisms in healthy and disease states.

## Materials and methods
### Experimental design, cell culture, and viral infections

Experiments were performed in human foreskin fibroblasts (HS68 cells). ATAC-seq experiments were also performed in human retinal epithelial cells (ARPE-19 cells). HS68 and ARPE-19 cell lines were

obtained from ATCC. Identity of cells was confirmed by whole-genome sequencing. All cells were routinely tested for mycoplasma contamination, and all tests returned negative. HCMV infections were performed using the TB40/E clinical isolate. Fibroblasts were infected at an MOI of 5 and cells were harvested 48 hr post-infection for RNA-seq, ATAC-seq, ChIP-seq, and HiChIP (*Figure 1*). Retinal epithelial cells were infected at an MOI of 10. The rate of infection was monitored by both FACS and microscopy. Exclusion criteria were pre-established in that events that were not detected in all replicates in each biological group were excluded from the differential analyses.

## Gene expression (RNA-seq)

Total RNA was isolated from $5 \times 10^6$ cells, either uninfected or infected (48 hr), using the RNeasy RNA Isolation kit (QIAGEN# 74104) as per instructions provided by the manufacturer. A total of 500 ng of RNA per sample was used as input material for ribosomal-RNA-depleted RNA-sequencing. First, ribosomal RNA (rRNA) was removed, and rRNA-free residue was cleaned up by ethanol precipitation. Subsequently, sequencing libraries were generated using the rRNA-depleted RNA by the Directional RNA Library Prep Kit. Briefly, fragmentation was carried out using divalent cations under elevated temperature in first strand synthesis reaction buffer (5X). First strand cDNA was synthesized using random hexamer primer and M-MuLV Reverse Transcriptase (RNaseH-). Second strand cDNA synthesis was subsequently performed using DNA Polymerase I and RNase H. In the reaction buffer, dNTPs with dTTP were replaced by dUTP. Remaining overhangs were converted into blunt ends via exonuclease/polymerase activities. After adenylation of 3′ ends of DNA fragments, adaptors with hairpin loop structure were ligated to prepare for hybridization. In order to select cDNA fragments of preferentially 150–200 bp in length, the library fragments were purified with the AMPure XP system (Beckman Coulter, Beverly, USA). Then 3 µl USER Enzyme (NEB, Ipswich, MA, USA) was used with size-selected, adaptor-ligated cDNA at 37°C for 15 min followed by 5 min at 95°C before PCR. PCR was performed with Phusion High-Fidelity DNA polymerase, Universal PCR primers, and Index Primer. Finally, products were purified (AMPure XP system) and library quality was assessed on an Agilent Bioanalyzer 2100 system (Agilent Technologies, Inc, Santa Clara, CA, USA). The libraries were sequenced on an Illumina NovaSeq 6000 at Novogene (paired-end, 150 bp read length).

RNA sequencing data were processed using the nf-core/rnaseq pipeline (version 3.8.1) (*Di Tommaso et al., 2017*; *Ewels et al., 2020*; *Harshil et al., 2022*). Initial quality control of the raw sequencing data was performed using FastQC (version 0.11.9) (*Andrews, 2010*). Low-quality bases and adapter sequences were then trimmed and filtered from the reads using Cutadapt (version 3.4) (*Martin, 2011*) and Trim Galore (version 0.6.7) (*Krueger et al., 2021*), respectively. rRNA sequences were subsequently removed from the aligned data using SortMeRNA (version 4.3.4) (*Kopylova et al., 2012*) to eliminate potential contamination from non-target RNA species. STAR aligner (version 2.7.10a) (*Dobin et al., 2013*) was used to align the trimmed reads to a custom reference genome of hg19 and Human herpesvirus 5 strain TB40/E clone TB40-BAC4 sequence (GenBank: EF999921.1). The resulting alignments were sorted and indexed using SAMtools (version 1.15.1). To further evaluate the quality of the RNA sequencing data, several tools were employed, including RseqQC (version 3.0.1) (*Wang et al., 2012*), Qualimap (version 2.2.2-dev) (*Okonechnikov et al., 2016*), dupRadar (version 1.18.0) (*Sayols et al., 2016*), and Preseq (version 3.1.1) (*Daley and Smith, 2013*). These tools provide comprehensive assessments of various quality metrics, such as read distribution, GC content, duplication rates, and library complexity, ensuring reliable data for downstream analysis. For transcript quantification, Salmon (version 1.5.2) (*Patro et al., 2017*) was utilized to estimate the expression levels of transcripts. Finally, differential gene expression analysis was performed using DESeq2 (version 1.30.1) (*Love et al., 2014*). Genes were considered differentially expressed if they had a twofold change and an adjusted p-value threshold of less than 0.01. Pathway enrichment analyses for the differentially expressed genes were performed using Enrichr (*Chen et al., 2013*; *Kuleshov et al., 2016*). An adjusted p-value threshold of 0.05 was used to generate the pathway enrichment figure (*Figure 5B*).

## Alternative splicing analysis

Exon–exon and exon–intron spanning reads were identified with the software AltAnalyze (version 2.1.4) using the Ensembl-72 human database, along with splicing event calculation and annotation with the MultiPath-PSI algorithm (see http://altanalyze.readthedocs.io/en/latest/Algorithms for algorithm details and benchmarking). To detect high-confidence splicing changes between HCMV-infected

cells and uninfected cells, splicing events that were not detected in all replicates in each biological group were excluded from the differential splicing analysis. Significant splicing changes were defined as splicing events with a change in percent spliced in PSI value between the two groups >10% (ΔPSI > |0.1|), with p-value <0.01.

## Chromatin accessibility (ATAC-seq)

Omni-ATAC-Seq was performed in replicates as previously published (*Corces et al., 2017*). Briefly, HCMV-infected and uninfected control cells were harvested by trypsinization for 5 min. Approximately 70,000 cells were transferred to a microfuge tube and washed once with PBS. The cell pellet was resuspended in 50 µl of ice-cold lysis buffer (10 mM Tris-HCl [pH 7.5], 10 mM NaCl, 3 mM $MgCl_2$, 0.1% NP-40, 0.1% Tween-20, and 0.01% Digitonin) and incubated on ice for 3 min, followed by centrifugation at $500 \times g$ for 5 min at 4°C. For the transposition reaction, nuclei were resuspended in 50 µl of Nextera transposition reaction mix consisting of 25 µl 2x TD Buffer, 2.5 µl Nextera Tn5 transposase (Illumina # FC-121-1030), and 22.5 µl of nuclease-free water. The reaction mixture was mixed 5–10 times by gentle pipetting and incubated at 37°C for 45 min. The transposed DNA was then purified by QIAGEN MinElute kit (QIAGEN # 28004) and eluted in 11 µl of elution buffer. 1 µl of the eluted DNA was used to assess quality on an Agilent Tapestation 4510. For library preparation, 10 µl of tagmented DNA was PCR amplified for 14 cycles in a 50-µl reaction volume using NEBNext High-Fidelity 2X PCR Master Mix (NEB # M0541S) and Nextera primers. The amplified library was purified and size selected using AMPure XP beads (Beckman # 63880) in a two-step protocol. First, DNA fragments >1000 bp were removed from the PCR reaction mixture by adding 22.5 µl (0.5x volume) of AMPure XP beads followed by incubation at room temperature (RT) for 10 min. The beads were magnetized, and the supernatant was transferred to a new tube. In the second size selection step, 58.5 µl (1.2x volume) of AMPure XP beads were added to the supernatant, mixed by pipetting, and incubated for 10 min at RT. Beads were washed twice with 80% ethanol, air-dried, and DNA eluted in 30 µl of 10 mM Tris-HCl [pH 8.0]. Library DNA concentrations and quality were assessed by a Qubit dsDNA HS assay kit (Q32854) and Agilent 4510 TapeStation system, respectively. Libraries were sequenced on an Illumina NovaSeq 6000 at the Cincinnati Children's Hospital Medical Center Genomics Sequencing Facility (paired-end, 150 bp read length).

ATAC-seq data were processed and aligned to the hg19 genome using the ENCODE ATAC-seq pipeline (V2.0.0) (*The ENCODE Project Consortium, 2012*; *Hitz et al., 2023*; *Lee et al., 2021a*; *Luo et al., 2020*). Peaks were called within the pipeline using MACS2 (*Zhang et al., 2008*). Differential chromatin accessibility analysis was performed using DiffBind 3.6.5 (*Rory Stark, 2017*; *Ross-Innes et al., 2012*) in R 4.2.1 (*R Development Core Team, 2023*). The 'conservative overlap' peak sets generated from the individual replicates by the ENCODE ATAC-seq pipeline were used for differential analysis. Peaks were considered to be differentially accessible if the FDR was less than 0.01 and the fold change was twofold or greater. A modified version of HOMER (*Heinz et al., 2010*) using a log base 2 likelihood scoring system was used to calculate motif enrichment statistics for a large library of human position weight matrix binding site models contained in build 2.0 of the CisBP database (*Weirauch et al., 2014*). DeepTools (v2.0.0) (*Ramírez et al., 2016*) was used to generate heatmaps of signal tracks across differentially accessible chromatin for each set of comparisons.

## Chromatin immunoprecipitation sequencing

$2 \times 10^7$ cells were seeded in 150 mm culture dishes and were either infected with HCMV at an MOI of 5 or kept uninfected. After 48 hr, the medium was removed, and cells were harvested by trypsinization followed by two washes with PBS. Cells were crosslinked with 10 ml of formaldehyde crosslinking solution (1x PBS, and 1% formaldehyde) for 10 min at RT. The crosslinking reaction was quenched by adding 2.5 M glycine to a final concentration of 125 mM and incubated for 5 min at RT. Cells were washed twice with ice-cold 1x PBS. Crosslinked cells were then transferred to a microfuge tube and frozen at −80°C until further use.

For chromatin preparation, crosslinked cells were resuspended in L1 buffer (50 mM HEPES-KOH [pH 8.0], 140 mM NaCl, 1 mM EDTA, 10% glycerol, 0.5% NP-40, 0.25% Triton X-100, and 1x protease inhibitors) and incubated at 4°C on a rotator for 10 min. Nuclei were pelleted by centrifugation and resuspended in L2 buffer (10 mM Tris-HCl [pH 8.0], 200 mM NaCl, 0.5 mM EGTA, and 1x protease inhibitors) with rotation for 10 min at RT. Isolated nuclei were then resuspended in sonication buffer

solution (10 mM Tris-HCl, 1 mM EDTA, and 0.1% SDS). To obtain chromatin fragments of 200–500 bp length, nuclei were sonicated using an S220 ultrasonicator (Covaris, LLC, Woburn, MA, USA) at 10% duty cycle, 175 peak power, 200 burst/cycle for 7 min at 4°C. A portion of the sonicated chromatin was run on an agarose gel to verify fragment sizes. The chromatin solution was centrifuged to pellet the debris, and the supernatant was collected in a fresh microfuge tube. Since the sonicated chromatin solution is devoid of any detergents or salts, the following were added at indicated final concentrations: Triton X-100 (1%), sodium deoxycholate (0.1%), glycerol (5%), NaCl (150 Mm), and protease inhibitor cocktail (1x). Chromatin was pre-cleared with Protein G Dynabeads for 45 min at 4°C on a rotator. Chromatin immunoprecipitations were performed in an SX-8X IP-Star Compact automation system (Diagenode LLC, Denville, NJ, USA) with 200 µl of pre-cleared chromatin (approximately 3 million cells), 21 µl of Protein A or G Dynabeads (Thermo Fisher Scientific, Waltham, MA, USA), and 5 µg of antibody. The following antibodies were used for ChIP: CTCF (ActiveMotif # 61311, lot 17118005), TEAD1 (ActiveMotif # 61643, lot 34614001), and H3K27ac (Abcam # ab4729, lot GR3357415-1). Immunoprecipitation of chromatin was carried out for 8 hr, after which the Dynabeads were sequentially washed with Wash Buffer 1 (10 mM Tris-HCl [pH 8.0], 150 mM NaCl, 1 mM EDTA, 0.1% SDS, 0.1% sodium deoxycholate, and 1% Triton X-100), Wash Buffer 2 (10 mM Tris-HCl [pH 8.0], 250 mM NaCl, 1 mM EDTA, 0.1% SDS, 0.1% sodium deoxycholate, and 1% Triton X-100), Wash Buffer 3 (50 mM Tris-HCl [pH 8.0], 2 mM EDTA, and 0.2% N-lauroylsarcosine sodium salt), and Wash Buffer 4 (TE + 0.2% Triton X-100) for 5 min each. For library preparation, immunoprecipitated chromatin was eluted in elution buffer (1x TE, 250 mM NaCl, and 0.3% SDS). Chromatin proteins and RNA were digested with Proteinase-K and RNase A, respectively, and DNA purified with the QIAGEN MinElute kit. A ChIP-DNA library was constructed using the NEBNext Ultra II DNA library preparation kit (E7645S) as per the manufacturer's instructions and purified with 0.6x volume of AMPure XP beads. Purified library DNA quality and quantity were assessed by an Agilent TapeStation 4150 and sequenced using an Illumina NovaSeq 6000 at the Cincinnati Children's Hospital Medical Center Genomics Sequencing Facility (Single end, 100 bp read length).

ChIP-seq data were processed and aligned to the hg19 genome using the ENCODE ChIP-seq pipeline (V2.0.0) (*The ENCODE Project Consortium, 2012*; *Hitz et al., 2023*; *Lee et al., 2021b*; *Luo et al., 2020*). Peaks were called within the pipeline using MACS2 (*Zhang et al., 2008*). Differential peak analysis was performed using DiffBind 3.6.5 (*Rory Stark, 2017*; *Ross-Innes et al., 2012*) in R 4.2.1 (*R Development Core Team, 2023*). The 'conservative overlap' peak sets generated from the individual replicates by the ENCODE ChIP-seq pipeline were used for differential analysis. Peaks were considered differentially enriched if the FDR was less than 0.01 and the fold change was twofold or greater.

## Chromatin looping interactions (HiChIP)

HiChIP libraries were prepared following the protocol from *Mumbach et al., 2016*. Experiments were performed using biological triplicates in both uninfected and infected cells. In brief, 10 million cells were crosslinked with 1% formaldehyde, followed by a cell lysis with Hi-C lysis buffer (10 mM Tris-HCl [pH 8.0], 10 mM NaCl, 0.2% NP-40, 1x Roche protease inhibitors) for 30 min at 4°C. Following cell lysis, 375 U of MboI (NEB, R0147) was used to digest, in situ, the crosslinked chromatin for 2 hr at 37°C. After filling-in the DNA ends by biotin-dTAP (Thermo # 19524016), dCTP, dGTP, and dTTP with 5 U/µl DNA polymerase (NEB # M0210), the DNA was ligated by T4 DNA ligase (NEB # M0202) at 4°C overnight. The ligated DNA was sonicated using a Covaris S220 platform (Covaris, LLC, Woburn, MA, USA) in nuclear lysis buffer (50 mM Tris-HCl [pH 7.5], 10 mM NaCl, 0.2% NP-40, 1x Roche protease inhibitors) at 4°C. The fragmented DNA was then diluted 10 times with ChIP dilution buffer (0.01% SDS, 1.1% Triton X-100, 1.2 mM EDTA, 16.7 mM Tris-HCl [pH 7.5], 167 mM NaCl), and the samples were pre-cleared with Protein A Dynabeads at 4°C for 1 hr, followed by immunoprecipitation with 8 µl H3K27Ac antibody (Abcam # ab4729, lot GR3374555-1) at 4°C overnight. DNA–protein complexes were captured by Protein A beads with rotation at 4°C for 2 hr. The Protein A beads were then washed sequentially with low salt, high salt, and LiCl wash buffer, followed by elution twice with 100 µl freshly prepared DNA elution buffer at 37°C (50 mM NaHCO$_3$, 1% SDS). The eluted chromatin was reverse crosslinked and purified by a PCR purification kit (QIAGEN). 5 µl of Streptavidin C-1 beads (Thermo Fisher # 65001) was used to capture the Biotin dATP labeled DNA. The captured DNA was transposed with 2.5 µl Tn5 transposase (Illumina). The beads were then sequentially washed by Tween wash buffer (5 mM Tris-HCl

[pH 7.5], 0.5 mM EDTA, 1 M NaCl, 0.05% Tween-20), 50 mM EDTA, and 10 mM Tris. After washing, PCR was performed by resuspending the beads with 23 μl $H_2O$, 25 μl 2X Phusion HF (New England Biosciences), 1 μl Nextera forward primer (Ad1_noMX), and 1 μl Nextera reverse primer (Ad2.X) at 12.5 μM. The PCR was run at eight cycles of (1) 72°C for 5 min, (2) 98°C for 1 min, (3) 98°C for 15 s, (4) 63°C for 30 s, followed by an extension at 72°C for 1 min. The post-PCR size selection was performed by two-sided selection with AMPure XP beads to capture the fragments between 300 and 700 bp. Samples were then sequenced on an Illumina NovaSeq 6000 at Novogene (paired-end, 150 bp read length).

HiC-Pro (version 2.11.4) was used to align and filter read pairs and identify the contact map (*Servant et al., 2015*). MboI restriction sites and default parameters were used to align the reads to the hg19 genome. After alignment, the read pairs were filtered to remove those that mapped to multiple locations, were not in a valid orientation, or were duplicated. Quality control metrics were generated at each step. HiC-Pro also reports the number of *trans* pairs, short-range *cis* pairs, and long-range *cis* pairs, as well as the number of pairs in each orientation (*Supplementary file 4*). The overlap conservative peaks generated by the ENCODE pipeline for the H3K27ac ChIP-seq data were used as input for calling loops using FitHiChIP (version 11.0) (*Bhattacharyya et al., 2019*). The parameters for FitHiChIP were the following: interaction type of 'peak to all'; bin size of 10 kb; lower loop distance of 20 kb; upper loop distance of 20 mb; background model of loose; bias correction of coverage bias regression; merge filtering enabled; and FDR of 0.01 (*Supplementary file 3*). Peaks were called from the HiChIP data using HiChIP-Peaks (version 0.1.2) (*Shi et al., 2020*) using the MboI restriction sites and an FDR of 0.01. The peaks were compared to the H3K27ac ChIP-seq peaks using bedtools (version 2.30.0) (*Quinlan and Hall, 2010*; *Supplementary file 4*). We observed strong agreement between experimental replicates (*Figure 2—figure supplement 2*), so we pooled the reads for the replicates and analyzed the resulting data as described above (*Supplementary files 3 and 4*). We identified shared loops by intersecting the coordinates of the anchors using a 5-kb padding. Specifically, we performed two intersections (one for the left anchors and one for the right anchors) using the bedtools window command with a -w parameter of 5000. Loops that intersected at both anchors were classified as shared (*Supplementary file 3*). We used the merged loops from the combined replicates as input. Replicate comparisons (*Figure 2—figure supplement 2*) were performed using the multiBamSummary and plotCorrelation programs from the deepTools software package (*Ramírez et al., 2016*). The parameters used for multiBamSummary were `--binSize` 10000 and `--distanceBetweenBins  0`. The parameters used for plotCorrelation were `--corMethod` pearson, `--skipZeros`, and `-- removeOutliers`.

## Assessment of protein expression levels (Western blots)

Uninfected and HCMV-infected cells were harvested, in triplicates, by trypsinization and washed twice with PBS. The cell pellet was then resuspended in 1x RIPA buffer supplemented with Halt Protease inhibitor cocktail (Sigma) and kept on ice for 30 min. Cell lysates were cleared by centrifugation at 12,000 RPM for 10 min and supernatant was collected in a new tube. Protein concentration was measured by the BCA method and 30 μg of protein loaded on 4–12% Bis-tris Novex gel (Invitrogen). After electrophoresis, proteins were transferred onto a Nylon membrane using Invitrogen's semi-dry transfer method. Proteins were blocked with Intercept blocking buffer (LI-COR) and membrane incubated with HCMV IE1/2 (Millipore # MAB810R), TEAD1 (Cell Signaling # 12292), YAP1 (Cell Signaling # 14074), pYAP1 (Phospho-YAP1; Ser127; Cell Signaling # 4911), THBS1 (Novus Biologicals # NB100-2059), CCN1 (Novus Biologicals # NB100-356), H3K27ac (Abcam # ab4729), β-Actin (Cell Signaling # 8457), or GAPDH (Invitrogen # ma5-15738) antibodies at 4°C for overnight. Following antibody binding, the membrane was washed 3 times with 1x TBST and incubated either with IRDye 800CW or IRDye 680RD conjugated secondary antibodies for 45 min at RT. For target protein quantification against a loading control (GAPDH and β-Actin), secondary antibodies against both target protein and loading control were added to the blot. The membrane was washed three times with 1x TBST and imaged on an Odyssey Dlx system (LI-COR, Biosciences LLC, Lincoln, NE, USA). Scanned images were then used for target protein quantification using Emperia Studio software (LI-COR, Biosciences LLC, Lincoln, NE, USA).

## Assessment of YAP1 localization

Uninfected and HCMV-infected cells were harvested by trypsinization and washed twice with PBS. Cytosolic and nuclear protein extracts were prepared using Pierce NE-PER Nuclear and Cytoplasmic

Extraction Reagent Kit (Thermo Scientific # 78833), as per the manufacturer's instructions. Protein concentration was measured by the BCA method and 15 µg of protein loaded on 4–12% Bis-tris Novex gel (Thermo Fisher Scientific, Waltham, MA, USA). Western blotting and imaging were performed as given above with the following antibodies: YAP1 (Cell Signaling # 14074), pYAP1 (Phospho-YAP1; Ser127; Cell Signaling # 4911), Histone H3 (Abcam # ab1791), and GAPDH (Invitrogen # ma5-15738).

### Validation of TEAD1 splicing (RT-PCR)

Total RNA was extracted from uninfected and HCMV-infected fibroblasts 48 hpi using the RNA mini kit from QIAGEN as per the manufacturer's instructions. 2.5 µg of total RNA was reverse transcribed using SuperScript IV VILO Master Mix (Thermo Fisher Scientific, Waltham, MA, USA) and 3 µl of cDNA was used in PCR to amplify the regions immediately flanking TEAD1 exon 6 using forward and reverse primers. Forward primer: 5′-ATCTCGTGATTTTCATTCCAAGC. Reverse primer: 5′-TGAGGACATGGC CGCCATGTGC.

### Data visualization

Visualization tracks for each dataset were created for the UCSC Genome Browser (*Kent et al., 2002*). Signal tracks (in bigWig format) were created using the bamCoverage program from the deepTools software package (*Ramírez et al., 2016*) with the parameters `--normalize`Using BPM and `--binSize` 10. For the RNA-seq signal tracks, the command was run twice in order to maintain strandedness. For the forward strand track, the parameter `--filterRNAstrand` forward was included. For the reverse strand track, the parameters `--filterRNAstrand` reverse and `--scale-Factor` –1 were included. The .hic files for the HiChIP data were created using the hicpro2juicebox. sh script from HiC-Pro (version 2.11.4) (*Servant et al., 2015*). Juicer (version 1.22.01) (*Durand et al., 2016*) was used for KR normalization.

### TEAD1-binding event phenotype enrichment analysis

To determine the significance of the overlap between TEAD1-binding events and GWAS-derived disease- and phenotype-associated genetic variants, we generated a custom GWAS catalog for each phenotype in an ancestry-specific manner. To this end, we downloaded the Genome Wide Association Studies Catalogue (https://www.ebi.ac.uk/gwas/) v1.0.2, as queried on June 20, 2019. Independent risk loci for each disease/phenotype were identified based on linkage disequilibrium (LD) pruning ($r^2 < 0.2$). Risk loci across these independent genetic risk variants were identified by LD expansion ($r^2 > 0.8$) based on 1000 Genomes Data using PLINK (v.1.90b). This created a list of disease risk loci, along with the corresponding genetic variants within the LD block. This list of variants was then used for RELI analyses as previously published (*Harley et al., 2018*). Phenotypes were considered 'TEAD1 loss specific' if: (1) they had three or more overlaps with TEAD1-binding event losses; (2) this overlap was significant according to RELI (corrected p < 0.01); (3) this overlap was not significant for unchanged TEAD1 peaks (p > 0.01); and (4) TEAD1-binding loss events were at least twofold enriched according to RELI. The three phenotypes meeting these criteria are presented in the Results, with further data included in *Supplementary file 14*.

## Acknowledgements

This research was funded by National Institutes of Health (NIH) R01 HG010730, R01 GM055479, and U01 AI130830 to MTW; R01 AR073228, R01 NS099068, and R01 AI024717 to MTW and LCK; R01 AI148276, U01 HG011172, U19 AI070235, and P30 AR070549 to LCK; T32 ES007250 to AVH; R21 DE026267 and R01 AI121028 to WEM; F32 AI172329 and T32 AI007245 to LAM-N; R01 AI164709 to BEG; US Department of Veterans Affairs, I01 BX003850 to KMK; US Department of Veterans Affairs, I01 BX001834 and I01 BX006254 to JBH. We would like to thank Jaimie Miser for her assistance with culture work, Phillip Dexheimer for his help with the virus expression database (https://vexd.cchmc.org/), the Cincinnati Children's Hospital Medical Center Genomics Sequencing Facility (RRID:SCR_022630) for high-throughput sequencing, the support of the Informatics Shared Facility in Information Services for Research (IS4R) at Cincinnati Children's Hospital Medical Center (RRID:SCR_022622), and Kevin Ernst for help with database management and computational support.

## Additional information

### Funding

| Funder | Grant reference number | Author |
| --- | --- | --- |
| National Institutes of Health | R01 HG010730 | Leah C Kottyan<br>Matthew T Weirauch |
| National Institutes of Health | R01 GM055479 | Matthew T Weirauch |
| National Institutes of Health | U01 AI130830 | Matthew T Weirauch |
| National Institutes of Health | R01 AR073228 | Leah C Kottyan<br>Matthew T Weirauch |
| National Institutes of Health | R01 NS099068 | Leah C Kottyan<br>Matthew T Weirauch |
| National Institutes of Health | R01 AI024717 | Leah C Kottyan<br>Matthew T Weirauch |
| National Institutes of Health | R01 AI148276 | Leah C Kottyan |
| National Institutes of Health | U01 HG011172 | Leah C Kottyan |
| National Institutes of Health | U19 AI070235 | Leah C Kottyan |
| National Institutes of Health | P30 AR070549 | Leah C Kottyan |
| National Institutes of Health | T32 ES007250 | Andrew VonHandorf |
| National Institutes of Health | R21 DE026267 | William E Miller |
| National Institutes of Health | R01 AI121028 | William E Miller |
| National Institutes of Health | F32 AI172329 | Laura A Murray-Nerger |
| National Institutes of Health | T32 AI007245 | Laura A Murray-Nerger |
| National Institutes of Health | R01 AI164709 | Ben E Gewurz |
| U.S. Department of Veterans Affairs | I01 BX003850 | Kenneth M Kaufman |
| U.S. Department of Veterans Affairs | I01 BX001834 | John B Harley |
| U.S. Department of Veterans Affairs | I01 BX006254 | John B Harley |

The funders had no role in study design, data collection, and interpretation, or the decision to submit the work for publication.

### Author contributions

Khund Sayeed, Investigation, Methodology, Writing – original draft, Writing – review and editing; Sreeja Parameswaran, Formal analysis, Investigation, Writing – original draft, Writing – review and editing; Matthew J Beucler, Omer A Donmez, Scott Richards, Hayley K Hesse, Sydney H Jones, Katelyn A Dunn, Jay Wright, Laura A Murray-Nerger, Investigation; Lee E Edsall, Data curation, Formal analysis, Writing – review and editing; Andrew VonHandorf, Audrey Crowther, Data curation, Formal analysis; Matthew R Hass, Supervision, Investigation, Visualization, Writing – review and editing; Carmy R Forney, Supervision, Investigation, Methodology, Project administration; Merrin Man Long

Leong, Investigation, Methodology; Vijay Yechoor, Conceptualization, Resources, Writing – review and editing; Ben E Gewurz, Supervision, Funding acquisition, Writing – review and editing; Kenneth M Kaufman, Formal analysis, Supervision, Investigation; John B Harley, Conceptualization, Supervision, Project administration, Writing – review and editing; Bo Zhao, Resources, Supervision, Investigation, Methodology, Writing – review and editing; William E Miller, Conceptualization, Supervision, Funding acquisition, Investigation, Writing – original draft, Writing – review and editing; Leah C Kottyan, Conceptualization, Resources, Data curation, Supervision, Funding acquisition, Visualization, Writing – original draft, Project administration, Writing – review and editing; Matthew T Weirauch, Conceptualization, Resources, Data curation, Software, Formal analysis, Supervision, Funding acquisition, Visualization, Writing – original draft, Project administration, Writing – review and editing

### Author ORCIDs
Sreeja Parameswaran http://orcid.org/0009-0002-3631-3669
Lee E Edsall https://orcid.org/0000-0002-0326-2829
Sydney H Jones https://orcid.org/0009-0005-2674-4029
Vijay Yechoor https://orcid.org/0000-0002-9981-6784
Bo Zhao https://orcid.org/0000-0002-8612-5597
Matthew T Weirauch https://orcid.org/0000-0001-7977-9122

Reviewer #1 (Public review): https://doi.org/10.7554/eLife.101578.3.sa1
Reviewer #2 (Public review): https://doi.org/10.7554/eLife.101578.3.sa2
Author response https://doi.org/10.7554/eLife.101578.3.sa3

---

# Additional files

### Supplementary files
Supplementary file 1. ATAC-seq data QC results.

Supplementary file 2. ATAC-seq differential peak analysis results.

Supplementary file 3. Chromatin looping interaction statistics.

Supplementary file 4. HiChIP data QC results.

Supplementary file 5. Chromatin looping event coordinates.

Supplementary file 6. Pairwise statistical comparison of datasets generated in this study using RELI.

Supplementary file 7. ChIP-seq data QC results.

Supplementary file 8. ChIP-seq differential peak analysis results.

Supplementary file 9. RNA-seq data QC results.

Supplementary file 10. HCMV gene expression levels (TPMs).

Supplementary file 11. RNA-seq DEG analysis results.

Supplementary file 12. Regions of intense TEAD1-binding activity loss near differentially expressed genes.

Supplementary file 13. Hippo pathway genes from KEGG.

Supplementary file 14. Phenotype-associated genetic variant overlap analysis results (RELI algorithm applied to GWAS).

MDAR checklist

### Data availability
All raw and processed sequencing data generated in this study have been submitted to the NCBI Gene Expression Omnibus (GEO; https://www.ncbi.nlm.nih.gov/geo/) under accession number GSE254741. A UCSC Genome Browser session for the hg19 genome build is available at http://genome.ucsc.edu/s/Ledsall/CMV_genomics. Data generated or analyzed during this study are included in the manuscript and supporting files.

The following dataset was generated:

| Author(s) | Year | Dataset title | Dataset URL | Database and Identifier |
|---|---|---|---|---|
| Weirauch M | 2024 | Human cytomegalovirus infection coopts chromatin organization to modulate TEAD1 transcription factor activity | https://www.ncbi.nlm.nih.gov/geo/query/acc.cgi?acc=GSE254741 | NCBI Gene Expression Omnibus, GSE254741 |

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
