## [Editor Report · eLife Assessment]

This interesting study presents **important** information on how human cytomegalovirus (HCMV) infection disrupts the activity of the TEAD1 transcription factor, leading to widespread chromatin alterations. The strength of evidence in revised manuscript is **convincing**, and includes additional functional data teasing out how TEAD1-driven chromatin changes might influence HCMV replication. This work will be of interest to the virology, chromosome biology and transcriptional co-regulation fields.

---

## [Referee Report · Reviewer #1 (Public review)]

The manuscript by Sayeed et al. uses a comprehensive series of multi-omics approaches to demonstrate that late-stage human cytomegalovirus (HCMV) infection leads to a marked disruption of TEAD1 activity, a concomitant loss of TEAD1-DNA interactions, and extensive chromatin remodeling. The data are thoroughly presented and provide evidence for the role of TEAD1 in the cellular response to HCMV infection.

However, a key question remains unresolved: is the observed disruption of TEAD1 activity a direct consequence of HCMV infection, or could it be secondary to the broader innate antiviral response? In this respect, the study would benefit from more in-depth experiments that assess the effect of TEAD1 overexpression or knockdown/deletion on HCMV replication dynamics. The new data provided by the authors in Reviewer Response Figures 1 and 2 suggest that the presence of constitutively expressed TEAD1 does not substantially impact HCMV replication and gene expression as assessed at 72 and 96 hours post-infection. However, this does not discount the fact that HCMV infection induces significant TEAD1-related chromatin changes that may impact other cellular functions.

---

## [Referee Report · Reviewer #2 (Public review)]

Summary:

This work uses genomic and biochemical approaches for HCMV infection in human fibroblasts and retinal epithelial cell lines, followed by comparisons and some validations using strategies such as immunoblots. Based on these analyses, they propose several mechanisms that could contribute to the HCMV-induced diseases, including closing of TEAD1-occupying domains and reduced TEAD1 transcript and protein levels, decreased YAP1 and phospho-YAP1 levels, and exclusion of TEAD1 exon 6. Some functional assays, using over-expression of TEAD1, are provided.

Strengths:

The genomics experiments were done in duplicates and data analyses show good technical reproducibility. Data analyses are performed to show changes at the transcript and chromatin level changes, followed by some Western blot validations.

Weaknesses:

For readers who are outside the field, some clarifications of the system and design would be helpful.

---

## [Author Response]

The following is the authors’ response to the original reviews.

**Reviewer #1 (Public review):**
The manuscript by Sayeed et al. uses a comprehensive series of multi-omics approaches to demonstrate that late-stage human cytomegalovirus (HCMV) infection leads to a marked disruption of TEAD1 activity, a concomitant loss of TEAD1-DNA interactions, and extensive chromatin remodeling. The data are thoroughly presented and provide evidence for the role of TEAD1 in the cellular response to HCMV infection.However, a key question remains unresolved: is the observed disruption of TEAD1 activity a direct consequence of HCMV infection, or could it be secondary to the broader innate antiviral response? In this respect, the study would benefit from experiments that assess the effect of TEAD1 overexpression or knockdown/deletion on HCMV replication dynamics. Such functional assays could help delineate whether TEAD1 perturbation directly influences viral replication or is part of a downstream/indirect cellular response, providing deeper mechanistic insights.

To examine the effect of TEAD1 on HCMV, we performed an experiment in primary human foreskin fibroblasts (HFF) which were stably transduced with constitutive TEAD1. To constitutively express TEAD1, we cloned the open reading frame of TEAD1 into pLenti-puro (Plasmid #39481 from Addgene). We selected for transduced cells using puromycin. For these experiments, we first assessed two multiplicities of infection (MOI): 1 and 10 (Reviewer Response Figure 1). Based on the TEAD1 expression in these cells relative to non-transduced HFF cells, we performed HCMV infection experiments in cells transduced with TEAD1 lentivirus at an MOI of 1.

For infections, we used a version of HCMV in which the C terminus of the capsi-associated tegument protein pUL32 (pp150) is tagged by enhanced green fluorescent protein (GFP) (PMID: 15708994). This experimental design allowed us to assess the impact of constitutive TEAD1 expression on HCMV infection. GFP and immediate early protein expression levels were measured 48 hours after infection by flow cytometry.

After infecting parent cells (no constitutive TEAD1) and TEAD1 constitutively expressing cells with a GFP-positive HCMV at MOIs of 0.3 and 1, we identified equivalent GFP expression in the two conditions, indicating equivalent levels of HCMV infection 48 hours after initial infection (Reviewer Response Figure 1A). We also identified equivalent immediate early protein expression at 48 hours after infection, as measured both by percent positivity (Reviewer Response Figure 1B) and mean florescent intensity (Reviewer Response Figure 1C). At 96 hours with an MOI of 3, constitutive expression of TEAD1 led to a slight reduction in the expression of the HCMV proteins pp65 (encoded by UL83) and UL44 at 72 and 96 hours post initial infection (Reviewer Response Figure 1D). These results suggest that TEAD1 expression has minimal effects, if any, on the expression of these two late HCMV proteins in fibroblasts. Regulation of particular HCMV genes by TEAD1 is likely to be central for HCMV replication and reactivation in other specialized cell types relevant to viral pathogenesis and disease. However, definitive studies are beyond the scope of the current study.

**Author response image 1. sa3fig1:** Constitutive TEAD1 expression reduces expression of two HCMV late genes at 72 and 96 hours after infection. A-C. Primary human foreskin fibroblasts with and without constitutive TEAD1 expression were infected with pp150-GFP HCMV at a multiplicity of infection (MOI) of 0.3 or 1 and assessed 48 hours post infection. A. HCMV positive cells were quantified by measuring the percent of cells that were GFP positive. B. The percentages of immediate early (IE1/IE2) positive cells were quantified by flow cytometry. C. The mean florescence intensity of immediate early positive cells was quantified by flow cytometry. D. Primary human foreskin fibroblasts with and without constitutive TEAD1 expression were infected with pp150-GFP HCMV at an MOI of 1 and assessed by Western blot at various time point post infection. UL44 and pp65 are expressed late in the cascade of HCMV gene expression. TEAD1 expression levels and uncropped Westerns are provided in Supplemental Figure S8

Reviewer Response Methods:

Flow cytometric analysis of viral entry and spread using GFP expression and HCMV immediate early (IE) protein staining

Parental and TEAD1 transduced human foreskin fibroblasts were seeded into 12-well plates at 1.0 × 10^5^ cells per well and either mock infected or infected with pp150-GFP HCMV (PMID: 15708994) at MOIs of 0.3 or 1 on the same day. Cells were trypsinized at appropriate time points and then neutralized with complete medium. Cell suspensions were spun down at 500g for 5 minutes, and the cell pellet was fixed in 70% ethanol for 30 minutes. Following fixation, cells were permeabilized in phosphate-buffered saline (PBS) containing 0.5% bovine serum albumin (BSA) and 0.5% Tween 20 for 10 minutes at 4°C, pelleted, and then stained with IE1/IE2 antibody (mAb810-Alexa Fluor 488) diluted in PBS supplemented with 0.5% BSA for 2 hours. Cells were washed with PBS supplemented with 0.5% BSA–0.5% Tween 20 and then resuspended in PBS. Cells were analyzed using a flow cytometer (BD Biosciences). Infected cells were also trypsinized at appropriate time points, neutralized in the appropriate media, and directly analyzed for GFP positivity on the flow cytometer.

Western blot analyses of HCMV protein expression in infected cells with and without constitutive TEAD1 expression

TEAD1 transduced and parental human foreskin fibroblasts were seeded into 6-well cell culture plates at a density of 3.0 × 10^5^ cells per well and either mock infected or infected with pp150-GFP HCMV (PMID: 15708994) at an MOI of 1. Whole-cell lysates were collected at various time points post-infection, separated by SDS-PAGE, and transferred to nitrocellulose for Western blot analysis. Western blots were probed with the following primary antibodies: anti-IE1/IE2 (Chemicon), anti-UL44 (kind gift of John Shanley), anti-pp65 (Virusys Corporation), and cellular β-actin antibody (Bethyl Laboratories). Next, each blot was incubated with appropriate horseradish peroxidase-conjugated anti-rabbit or anti-mouse IgG secondary antibodies. Chemiluminescence was detected and quantified using a C-DiGit blot scanner from Li-Cor.

**Reviewer #2 (Public review):**
Summary:This work uses genomic and biochemical approaches for HCMV infection in human fibroblasts and retinal epithelial cell lines, followed by comparisons and some validations using strategies such as immunoblots. Based on these analyses, they propose several mechanisms that could contribute to the HCMV-induced diseases, including closing of TEAD1-occupying domains and reduced TEAD1 transcript and protein levels, decreased YAP1 and phospho-YAP1 levels, and exclusion of TEAD1 exon 6.Strengths:The genomics experiments were done in duplicates and data analyses show good technical reproducibility. Data analyses are performed to show changes at the transcript and chromatin level changes, followed by some Western blot validations.Weaknesses:This work, at the current stage, is quite correlative since no functional studies are done to show any causal links. For readers who are outside the field, some clarifications of the system and design need to be stated.
**Reviewer #2 (Recommendations for the authors):**
Here are some specific questions:(1) Since all current analyses are correlative, it is difficult to know which changes are of biological significance. For example, experiments manipulating TEAD transcription factor or YAP with effects on how cells respond to HCMV infection would significantly strengthen the conclusions, which are largely speculations now.

Please see response to Reviewer 1, which highlights newly added functional assays that include the constitutive (forced) expression of TEAD1, as suggested.

(2) How similar are these cell lines (human fibroblasts and retinal epithelial cell lines) resembling the actually infected cells in patients that lead to symptoms?

In infected cells in patients, HCMV initially infects both fibroblasts and epithelial cells. HCMV penetrates fibroblasts by fusion at the cell surface but is endocytosed into epithelial cells (PMID: 18077432). Thus, most experimental studies of HCMV in vitro use primary human foreskin fibroblasts and a retinal epithelial cell line, as we do in this study.

Additional information on primary human fibroblasts as a model of HCMV infection in humans

There is a nice review article that provides the history of the study of the molecular biology of HCMV that describes how Stanley Plotkin from the Wistar Institute first identified human fibroblast HCMV infected cells (PMID: 24639214). The primary fibroblasts of the foreskin of neonates are available commercially (sometimes called HS68) and model neonatal HCMV infection. Neonatal HCMV, or Congenital Cytomegalovirus, is a leading cause of congenital infection and a significant cause of non-genetic hearing loss in the US (https://www.cdc.gov/cytomegalovirus/congenital-infection/index.html). While many infected newborns appear healthy at birth, a substantial percentage experience long-term health problems, including hearing loss, developmental delays, and vision problems (PMID: 39070527).

More information on ARPE-3 as a model of HCMV infection in humans

HCMV retinitis is a leading cause of vision loss and results from HCMV infection of retinal cells. Retinal epithelial cells are the primary target for HCV infection in the eye. The cell line ARPE-19 is derived from a primary human adult retinal pigment epithelium explant and is commonly used to study HCMV and is thought to be physiologically relevant to the human infection (PMID: 8558129 and 28356702). When compared to primary retinal pigment epithelia, ARPE-19 cells develop a similar cellular and molecular phenotype to primary cells from adults and neonates (PMID: 28356702).

(3) What is the rationale for using 48 hours' infection? Is this the typical timeframe for patients to develop symptoms?

HCMV genes are expressed in a temporally controlled manner (PMID: 35417700). Early genes (within the first 4 hours) are involved in regulating transcription, while genes within 4-48 hours are involved in DNA replication and further transcriptional regulation. The 48 hour mark corresponds to the onset of significant viral replication and interactions between the virus and the host immune response. After 48 hours, late genes are expressed, which encode structural proteins as well as viral proteins that inhibit host anti-viral responses. Most studies that focus on the role of HCMV’s early and immediate early genes are performed at 24 or 48 hours. Similarly, most studies that assess the initial innate immune response to HCMV are performed within the initial 48 hours after in vitro infection.

In most people with healthy immune systems, there are no symptoms (PMID: 34168328). While 60% of people in developed countries and 90% of those in developing countries are serologically positive for past infection, it is challenging to study the kinetics of symptom development due to heterogeneity in the initial virion exposure, the cell types that are initially infected, and immune response. HCMV persists throughout the lifetime of the infected individual by establishing latent infection.

Also, among all these large-scale global changes, what are primary and what are secondary?

A kinetic study with many timepoints would be needed to identify the primary and secondary genomic changes associated with HCMV infection. These experiments, while exciting, are beyond the scope of this manuscript.

(4) Fig.2: In addition to the changes for each cell type, comparison of unchanged, closed and opened with infection regions between the two cell types could be informative for commonalities and differences between cell types.

This was a good suggestion. We have added a new Supplemental Figure S2, which compares the differentially accessible regions between the two cell types:

We have also added the following sentence to the Results section:

“Comparison of differentially accessible chromatin between ARPE and HFF revealed that the vast majority of the HCMV-induced changes are specific to one of the two cell types (Supplemental Figure S2).”

(5) "Of the 23,018 loops present in both infected and uninfected cells, only 10 are differential at a 2-fold cutoff and a false discovery rate (FDR) <0.01."

We thank the reviewer for drawing our attention to the differential chromatin looping analysis. Your comment prompted us to re-examine the methodologies we employed to identify differential chromatin looping events between uninfected and infected cells. In the process, we realized that the relatively low resolution of chromatin looping assays such as HiChIP might require additional care in classifying a particular loop as shared or differential when comparing two experimental conditions. We have thus revamped our differential chromatin looping methodologies by adding 5kb “pads” to either end of each chromatin loop “anchor”.

The corresponding passage now reads:

“We next used the HiChIP data to identify HCMV-dependent differential chromatin looping events (see Methods). In total, uninfected cells have 143,882 loops. With HCMV infection, 90,198 of these loops are lost, and 44,045 new loops are gained (Supplemental Dataset 3). Because the number of altered loops was large, we repeated loop calling and differential analysis with FDR values less than 0.05, 0.01, and 0.001 (Supplemental Dataset 3). For all three cutoffs, the percentage of loops specific to an infection state were very similar. We also randomly downsampled the number of input pairs used for calling loops to verify that our results were not due to a difference in read depth (Supplemental Dataset 3). For the three smaller subsets of data, the number of loops specific to an infection state only changed slightly. The full quantification of each chromatin looping event and comparisons of events between conditions are provided in Supplemental Dataset 6.”

Are these cells asynchronous and how to determine whether certain changes are not due to cell cycle stage differences?

Cells were plated to an identical density of cells per well before either mock or HCMV infection for this study. Based on the differentially expressed genes cell cycle pathways were not amongst the top 50 enriched molecular pathways.